# Reconfiguring Pain Interpretation Within a Social Model of Health Using a Simplified Version of Wilber’s All Quadrant All Levels Framework: An Integral Vision

**DOI:** 10.3390/bs15050703

**Published:** 2025-05-20

**Authors:** Mark I. Johnson

**Affiliations:** Centre for Pain Research, School of Health, Leeds Beckett University, City Campus, Leeds LS1 3HE, UK; m.johnson@leedsbeckett.ac.uk

**Keywords:** pain, biopsychosocial, all quadrant all level integral framework, Ken Wilber, painogenic environment, salutogenesis, health promotion

## Abstract

Despite the proliferation of biomedical and psychological treatments, the global burden of chronic intractable (long-term) pain remains high—a treatment-prevalence paradox. The biopsychosocial model, introduced in the 1970s, is central to strategies for managing pain, but has been criticised for being decontextualised and fragmented, compromising the effectiveness of healthcare pain support services and patient care. The aim of this study was to apply a simplified version of Ken Wilber’s All Quadrant All Levels (AQAL) framework to pain in a healthcare context to advance a biopsychosocial understanding. Utilising domain knowledge, the author mapped features of pain and coping to intrasubjective, intraobjective, intersubjective, and interobjective quadrants (perspectives), as well as levels of psychological development. Narratives were crafted to synthesize the findings of mapping with literature from diverse disciplines within the contexts of salutogenesis and a social model of health. The findings showed that AQAL-mapping enhanced contextual biopsychosocial coherence and exposed the conceptual error of reifying pain. Its utility lay in highlighting upstream influences of the painogenic environment, supporting the reconfiguration of pain within a social model of health, as exemplified by the UK’s Rethinking Pain Service. In conclusion, a simple version of the AQAL framework served as a heuristic device to develop an integral vision of pain, opening opportunities for health promotion solutions within a salutogenic context.

## 1. Introduction

There are concerns about the global burden of long-term (chronic) treatment-resistant (intractable) pain on individual suffering, societal well-being, and healthcare systems ([33]; [93]; [169]; [256]). Long-term pain affects about one-third of people globally and continues to rise, despite advances in pain science and ever-increasing varieties of biomedical and psychological treatment interventions—a treatment-prevalence paradox ([48]; [55]; [73]; [128]; [164]; [257]).

Major organisations, including the International Association for the Study of Pain (IASP) and the World Health Organisation (WHO) define pain as “An unpleasant sensory and emotional experience associated with, or resembling that associated with, actual or potential tissue damage”, ([169]). Centring definitions of pain on ‘tissue damage’ may contribute to tissue-centric myopia, a dominance of biomedical solutions, and damage-loaded warmongering pain language ([96]; [97], [102]; [144]; [184]).

In the modern world, medicine, as the dominant paradigm, may have diminished the cultural meaning of pain by reducing it to a representation of a damaged body that values objective measurement of the structure and function of damaged tissue, stripping pain of its deeper, subjective significance ([43]; [51]; [126]; [184]). Standardisation of treatment, datafication of health, and over-medicalisation of pain may narrow the existential meaning of pain and contribute, at least in part, to an ‘opioid crisis’ of drug misuse ([121]; [201]; [220]). The over-pathologisation of pain—treating it solely as a consequence of tissue damage—may inadvertently promote a cultural view in which pain is regarded only as an unwelcome disruption, devoid of any redemptive or transformative value, and thus something to be eliminated at all costs rather than embraced as an intrinsic aspect of the human experience ([27]; [97]; [184]; [201]; [224]).

People with long-term pain often have adversarial relationships with their healthcare providers due to miscommunication and misalignment of values, resulting in distrust and differing perspectives ([195]; [217]). Research on first-person experiences of people living with long-term pain suggests that these individuals face significant emotional, psychological, and social challenges, including a fragmented sense of self and relationships, feelings of isolation and frustration, and a lack of understanding from others, including healthcare practitioners ([214], [213]; [221]). Patients frequently feel their pain is not taken seriously and that unique cultural, social, and individual experiences that place pain in a broader existential context are overlooked ([148]; [195]; [214], [215], [216], [218], [217]).

In the 1970s, George Engel developed the biopsychosocial model to balance the biomedical dominance of health by integrating biological, psychological, and social factors in understanding and treating illness ([56]). This model was adapted for pain and endorsed by major organisations, including the IASP and the WHO, as the cornerstone to person-centred biopsychosocial pain management ([88], [90]; [251]). The biopsychosocial model attempts to overcome the limitations of biomedically focused approaches by integrating psychological and social factors for a more holistic understanding and treatment of pain and has been the cornerstone of contemporary pain understanding and management strategies since the 1980s ([56]; [175]).

In recent years, the biopsychosocial model of pain has come under scrutiny because biopsychosocial domains remain dislocated from each other resulting in fragmentation of healthcare, with practitioners struggling to address all dimensions of pain giving rise to disjointed, resource-intensive care plans that are biased towards the expertise of individual healthcare practitioners ([23]; [34]; [36]; [40]; [50]; [131]; [147]; [189]; [199]; [202]). The biopsychosocial model is decontextualised from *being* a human, with insensitivity to a person’s inner state, leading to calls for better integration of phenomenological elements into theories to acknowledge the subjective nature of pain ([34]; [199]).

The biopsychosocial model was developed within a Western, Educated, Industrialised, Rich, and Democratic (WEIRD) worldview. WEIRDness, as posited by Joseph Henrich with supporting empirical research ([78], [79], [80]; [81]; [82]; [138]), tends towards impersonal prosociality, individualism, analytical thinking, and judgments that prioritise personal attributes and intentions over relationships and context. The concept of WEIRDness has been criticised for oversimplifying complex historical processes and reinforcing biases of societal structures ([28]; [65]) but it helps to contextualise dilemmas arising from the biopsychosocial view of pain—that pain is an enemy to be removed ([184]).

In less WEIRD societies, pain is viewed holistically as interconnected with spiritual, emotional, and physical health ([3]; [151]). Discomfort from pain is accepted as a natural part of life, with community support playing a significant role in providing physical and emotional support through cultural practices and rituals ([10]; [63]; [180]). There is a paucity of research on WEIRD bias in pain. An observational study by [226] ([226]) found that clinical research serves as a benchmark for ‘objective truth’ to guide authoritative biomedical evaluations in WEIRD countries but does not align with the values and experiences of individuals from non-WEIRD contexts who prefer ‘indigenous treatments’ for the subjective reality of pain. This reveals discrepancies between the values people place on care for subjective symptoms and authoritative evidence-based assessments of treatments in the biomedical literature and highlights the need for broader perspectives that encompass Eastern and Western philosophies.

The holistic whole-person perspective of pain aligns with contemporary views of IASP and WHO, yet both organisations define an optimum human state as one that includes the absence of pain and discomfort. This has motivated, at least in part, the inclusion of chronic (long-term) primary pain as a disease entity in its own right in the most recent version of the International Classification of Diseases-version 11 (ICD-11) ([146]; [250]). Although this helps to justify policy decisions to resource healthcare services for people living with chronic (long-term) pain, it also serves the interests of pharmaceutical and medical device industries who provide biomedical treatments for pain ([107]; [228]; [233]). An unintended consequence of categorising pain as a disease entity in its own right may be a worsening of societal harm from the overuse and misuse of analgesic medications, contributing to rising overdose deaths involving both prescription and illicit opioids ([19]; [85]; [97]; [184]; [201]; [220]).

Other predicaments in pain medicine include conceptual issues and a lack of confidence in clinical research evidence. [32] ([32]) draw attention to conceptual issues, including misnomers (e.g., conflating nociception with pain, stimulus-response confusion), fallacies (e.g., reification of pain, mereological fallacy), and speculative constructs (e.g., pain sensitivity), highlighting the need to improve the precision and clarity of language, logic, and reason. Moreover, most systematic reviews of acute pain interventions show minimal clinically significant differences, and this lack of robust evidence extends to long-term pain interventions as well ([1]). Moore et al. argue that much clinical research in pain and anaesthesia is ‘flawed’ and ‘futile’ with an urgent need to improve quality and reliability ([134]). In fact some analytical philosophers have suggested that pain may be too complex and idiosyncratic to be explained in a generalised manner or targeted by medical interventions ([37]).

A further challenge is that the biopsychosocial model is ‘static’, failing to account for the dynamic nature of pain over brief and extended periods. A meta-ethnography by [125] ([125]) conceptualised persistent pain acceptance as a fluid journey interconnected with socio-cultural-political worlds, likening the experience of pain to an ecosystem with a complex interplay of biological, psychological, social, and environmental factors.

In today’s professional landscape, intellectual value is placed on specialised knowledge, such as expertise in the neurobiology of nociception, genetic factors influencing ‘pain sensitivity *[sic]*’, the psychology of pain, the pharmacology of analgesics, the efficacy of pain management therapies, or the socio-economic impacts of pain on public health. This emphasis on specialised knowledge has favoured depth and detail over breadth and context ([162]), making it challenging to represent pain as an integrated whole ([98]; [162]). [66] ([66]) recognises the challenge of pain and psychological integration, arguing that pain’s complexity is part of a broader issue in the scientific study of the mind.

Increasingly, socio-ecological and systems theory frameworks have been applied to visually represent the complex interplay of biological, psychological, and social influences of pain, most commonly in the context of improving care ([5]; [21]; [38]; [77]; [123]; [145]; [222]; [253]). These frameworks do not explore the complexity of pain from the perspective of psychological integration. Thus, to date, there has been no attempt to develop an integral understanding of the biopsychosocial nature of pain as a dynamic, subjective, evolving experience.

Ken Wilber (1949-) is a populist American philosopher who developed an All Quadrants All Levels (AQAL) integral framework by combining ideas from psychology, science, and Eastern and Western philosophy to create a holistic model of the ‘integral mind’—Wilber’s integral theory of consciousness. Wilber communicated this scholarship through a comprehensive series of books published over a 45-year period, some of which have been drawn upon in the present article ([235], [236], [237], [238], [239], [240], [241], [242], [243]). In general, Wilber’s scholarship bypassed formal academic scrutiny in peer review journals and lacks empirical validation, undermining the scientific credibility of Wilber’s ‘Integral Theory of Everything’ and Wilber’s ‘integral movement’. Criticisms of the AQAL framework are that in its full form it is overly complex, granular, and hierarchical ([22]; [42]; [76]; [136]; [150]; [161]).

Despite these criticisms, a simplified version of the AQAL framework may have potential as a simple heuristic device to explore pain. A simplified AQAL framework would incorporate Wilber’s four quadrant viewpoints that explore the following:individual inner experience (intrasubjective personal inner state—interior)collective inner cultural meanings (intersubjective shared inner understandings—interior)individual externally manifested bodily processes (intraobjective observable physiology and behaviour—exterior)collective external settings (interobjective shared social and environmental systems and structures—exterior)

These four quadrants (viewpoints) do not require validation; they simply exist. A simplified AQAL framework could also include Wilber’s gross levels (stages) of psychological development that reflect cognitive growth and evolving worldview of individuals and collectives, both accepted notions within mainstream psychology ([11]; [116]; [155]). Spirituality plays a central and evolving role in Wilber’s scholarship. In the context of this article, the term spirituality is employed in a broad, non-religious sense, denoting the ways in which individuals seek and articulate meaning and purpose in life (see Section 4.2.3).

To date, there has been no attempt to apply a simplified AQAL framework to pain. The unique features of the AQAL framework are its integration of individual–collective and interior–exterior dimensions with psychological development. The AQAL framework compares favourably with the Dahlgren & Whitehead socio-ecological framework ([39]), which does not account for subjective–objective dimensions or dynamic psychological changes, and Bronfenbrenner’s bioecological systems framework ([20]), which lacks multiple dimensions and psychological levels of reality. The AQAL framework has potential for a more nuanced, dynamic, and coherent understanding of pain, addressing its biopsychosocial nature as an evolving experience within the socio-ecological milieu of modern living.

### Aim

The aim of this study was to apply a simplified version of Wilber’s AQAL integral framework to pain in a healthcare context to explore its utility to uncover insights that advance a biopsychosocial understanding.

“Pain” was analysed as a single, unified entity (experience) rather than breaking it down into its various components (e.g., everyday pain, acute pain, chronic (long-term); or ICD-11 classifications; or nociceptive, neuropathic, nociplastic). Likewise, “healthcare” was considered in the broadest sense, without delineating its various components such as community, primary, secondary, or tertiary.

The overarching research question was “What insights can be gained by applying the AQAL framework to pain in a healthcare context?” The hypothesis posits that applying a simplified version of the AQAL framework to pain in a healthcare context can provide insights into the multifaceted nature of pain that advances a biopsychosocial understanding. The objectives are to leverage a simplified AQAL framework to:Enhance the biopsychosocial perspective of pain—integral painEnhance the perspective of holistic person-centred healthcare—integral healthcareHighlight implications for the field, including future directions for research.

The approach seeks to gain insights that contribute to the advancement of a biopsychosocial understanding of pain within the context of evolutionary-mismatch ([95]), the painogenic environment ([96]), and salutogenesis (i.e., the origins and resources that support health and well-being ([122]; [133]; [157])).

## 2. Materials and Methods

The methodological approach involved defining the research question, mapping the author’s domain knowledge to the AQAL framework, analysing the mapped data for patterns against published literature, and writing a narrative that reflected the author’s unique insights and experiences positioned within a broader body of scholarship on evolutionary-mismatch ([95]), the painogenic environment ([96]), and salutogenesis ([104]; [157]).

The author interpreted the basic elements of Wilber’s AQAL framework (Section The Author’s Interpretation of Wilber’s AQAL Framework) and then used domain knowledge to map the features of pain, coping strategies, and approaches to care to quadrants (Section 3.1) and levels of worldview and psychological development (Section 3.2). The features of pain were extracted into tables to facilitate comparison and analysis of data, from which text narratives were crafted supported by literature from various peer-reviewed sources such as review articles, research studies and books from a diverse array of disciplines, including biomedical sciences, psychology, philosophy, humanities, sociology, and ecology. The purpose of narratives was to illustrate, interpret, and apply the data mapped to the AQAL framework in the context of a pain patient, healthcare practitioner, and healthcare service (Section 3.3). The implications of the findings are discussed in relation to the study hypothesis to advance the understanding of the biopsychosocial perspective of pain (objective 1), holistic person-centred healthcare (objective 2), and future directions for practice and research (Section 4).

The author utilised contextual understanding, reflexivity, and holistic analysis to pursue rigorous accounts. The author is a UK non-clinical pain scientist with over 35 years of experience of undertaking quantitative and qualitative research and a philosophical positionality that aligns with post-positivist critical realism. The author advocates for holistic, person-centred healthcare, grounded in salutogenesis.

An interpretive reflexive approach was employed to explore emerging inquiries, constructing nuanced narratives for deeper insight into the meanings and implications. The author engaged in a continuous process of reflexivity by documenting essential reflections and triangulating domain knowledge with published literature and debriefings with a network of colleagues (pain scholars). This approach informed the iterative analyses and helped acknowledge and address potential biases or assumptions that could influence the interpretation of the data ([26]).

### The Author’s Interpretation of Wilber’s AQAL Framework

In 2020, Duffy published a useful primer on the application of the integral framework to mental health ([49]). The author created a simplified version of the AQAL framework using Duffy’s primer and a collection of Wilber’s published scholarship ([235], [236], [237], [238], [239], [240], [241], [242], [243]). The author’s interpretation of the basic elements of Wilber’s AQAL framework are summarised in Figure 1.

The author’s interpretation of broad areas of knowledge relevant to each quadrant is summarised in Figure 2.

The individual-interior quadrant (upper left—UL) focuses on subjective, firsthand experiences, such as emotions and thoughts, investigated using qualitative research methods. The individual-exterior quadrant (upper right—UR) focuses on an individual’s body and behaviours observable in the ‘exterior world’, investigated using measurable scientific methods. The collective-interior quadrant (lower left—LL) focuses on shared cultural values and collective meaning-making through social interactions, investigated using ethnography and cultural studies. The collective-exterior quadrant (lower right—LR) focuses on how systemic factors in the external environment shape reality and influence behaviours and conscious experiences, investigated using systems-based analysis and qualitative inquiry.

Wilber used sobriquets to assist understanding of each quadrant; “I” for the individual-interior quadrant (UL), “It” for the individual-exterior quadrant (UR), “We” for the collective-interior quadrant (LL), and “Its” for the collective-exterior quadrant (LR). Personalised sobriquets may include the following:UL (“I”)—“Me”, “My Inner State”, “My Self”, “My Psychology”;UR (“It”)—“My Body”, “My Behaviour”, “My Biology”;LL (“We”)—“My/Our Culture”, “My/Our Worldview”, “My/Our Story”;LR (“Its”)—“My/Our Settings”, “My/Our Society”, “My/Our Environment”.

Wilber contends that individuals and societies develop through levels of increasing psychological complexity for cognitive, emotional, moral and spiritual domains (i.e., various *lines* of development), and each new level envelops, preserves, and transcends the previous level, i.e., nested Holons ([238], [239], [243]). Examples of lines of development for each quadrant are provided in Figure 3 based on the author’s interpretation of Wilber’s descriptions of the AQAL framework ([239], [242]).

The author decided to explore the development of worldview (LL) and cognitive self (UL). Wilber based the levels of worldview on Jean Gebser’s five structures of collective consciousness. Jean Gebser (1905–1973) was a Swiss philosopher, linguist, and poet known for scholarship on the different ways humans perceive and interact with reality (e.g., *The Ever-Present Origin* ([69])). Wilber based the levels of psychological development on the scholarship of Jean Piaget and Robert Kegan ([111]; [179]).

Wilber’s AQAL framework also considers ‘states of consciousness’ reflecting degrees of awareness moment to moment, such as sleep, dreaming, waking, meditative, and spiritual states, and ‘types’ such as gender, ethnicity, culture, and tradition, and other typologies that influence how individuals experience and interpret their reality ([238], [239], [242]). States of consciousness impact an individual’s experience of pain moment to moment, while levels of development influence an individual’s long-term experience and understanding of pain, including their longer-term coping strategies ([163]). States of consciousness and types were not considered in the analysis, although the features that distinguish levels of development and states of consciousness were charted.

## 3. Results

In this section, information is mapped to the author’s interpretation of Wilber’s AQAL framework using the approach described in Section 2. Table 1 presents the author’s interpretation of Wilber’s description of the features of each quadrant primarily described in ([239], [242]).

General quadrant perspectives that arise from the author’s interpretation of Wilber’s AQAL framework are presented in Figure 4.

### 3.1. Mapping Pain to Quadrants

The result of mapping aspects of pain to each quadrant, based the author’s domain knowledge and interpretation of Wilber’s AQAL framework, is presented in Table 2, with further detailed mapping available as Appendix A.

The overarching perspectives of pain that emerge from mapping pain to quadrants are summarised in Figure 5.

The mapped data were reviewed and a narrative crafted, drawing on contemporary literature to illustrate and interpret the data in relation to each quadrant and study objectives.

#### 3.1.1. Individual-Interior: The Inner Experience of Pain

The individual-interior quadrant reflects the ‘inner world’ of the individual, i.e., the personal subjective experience of pain. The quadrant represents the mind-centric aspect of the biopsychosocial model, focusing on how pain impacts mental activities and the sense of self, influencing thoughts about health and quality of life.

The quadrant reflects the individual’s sensory, emotional, and cognitive aspects of pain such as location, intensity, discomfort, suffering, meaning, and relief; and the embodied nature of pain, recognising it as a bodily experience deeply connected to the sense of self. This quadrant acknowledges pain as an adaptive mental phenomenon that incorporates the body’s context through continuous ‘signal exchange’ between the physical body (individual-exterior), culture (collective-interior), and environmental systems (collective-exterior). The quadrant reflects mental qualities of a body in pain functioning within evolutionary limits (liminality) to take necessary actions (defence) and reduce uncertainty (active inference) ([30]; [205]; [206]).

Interventions targeting the individual-interior focus on the individual’s mental state to reduce the severity and quality of pain, improve emotional distress and modify pain appraisal to alleviate suffering and improve daily functions. A sample of approaches includes psychological techniques, (e.g., cognitive-behavioural therapy (CBT), acceptance and commitment therapy (ACT)), complementary therapies targeting ‘mindbody’ (e.g., mindfulness), engagement in art, and spiritual techniques. There is an extensive and robust body of knowledge about pain psychology and moderate certainty evidence that CBT has small or very small beneficial effects for reducing pain, disability, and distress in chronic (long-term) pain ([246]). There is growing phenomenological research on first-person pain experience in healthcare contexts (e.g., [217]). Future directions include exploring pain within the context of the inner self and engaging in creative and culturally adapted education to help individuals reconceptualise pain, reshape their sense of self, and find new ways to express their experiences (e.g., [100]; [211]).

#### 3.1.2. Individual-Exterior: Physiology and Behaviour

The individual-exterior quadrant represents the biological aspect of pain, focusing on bodily structures and functions, i.e., physiology and behaviour. It includes observable processes such as nociception, sensitisation, and neuroplasticity, as well as physiological stress responses, outward expression of the inner state of pain, and behavioural coping strategies. This quadrant reflects physiological processes related to actual or potential tissue damage (e.g., injury, disease), maladapted physiology (long-term central sensitisation, aberrant neuroplasticity), and observable behaviours (expression of pain, escape, guarding). It also involves adjustments to maintain homeostasis and allostasis in response to internal and external changes associated with being in pain.

Interventions targeting the individual-exterior quadrant focus on altering pathology to resolve and/or relieve pain and techniques that interact with physiology to modulate nociceptive processes. A sample of approaches includes tissue-targeting treatments, such as drugs, surgery, neuromodulation, massage, and electrophysical agents, as well as psychological interventions to modify behaviour. There is a vast body of in-depth knowledge on the bioscience of nociceptive processes associated with pain from a third-person perspective, driven in part by a search for objective biomarkers of pain ([54]). Biomedical treatments (both medical and non-medical) dominate pain management despite long-standing concerns about overmedicalisation ([97]; [201]). There is a crisis in confidence of the evidence base for biomedical treatments provided by clinical research on pain and anaesthesia ([134]). Future directions related to this quadrant include advancing knowledge towards individualised precision diagnoses and treatments ([52]).

#### 3.1.3. Collective-Interior: The Shared Culture of Pain

The collective-interior quadrant reflects shared societal beliefs, morals, values, and norms shaping the experience and expression of pain. It represents the often-neglected culture-centric aspect of the social domain of biopsychosocial pain. For individuals, this quadrant involves being part of a intersubjective milieu of shared beliefs and values about pain, expressed through narrative. This quadrant reflects mutual understanding of and culture towards pain.

In WEIRD societies, this ‘culture of pain’ is influenced by healthcare organisations through guidelines for clinical practice, e.g., the IASP ([89]), the WHO ([251]), and the UK’s Faculty of Pain Medicine ([57]). Beliefs, attitudes, values, meanings, and expectations about pain are deeply embedded, yet fluctuate according to context, e.g., in social versus clinical situations. Steiner & Miglio proposes a critical conception of the ‘intersubjective self with pain’ as an intricate, multi-layered phenomenon, deeply embedded in the constitution of one’s bodily self emerging from a web of intercorporeal, social, cultural, and political relations, providing insight to how structural conditions make experiences more or less painful ([196]). The dynamics of the clinical encounter takes place in the intersubjective space of the collective-interior quadrant. There have been calls for greater attention to reframing the clinical encounter by considering intersubjectivity, empathy, prospection, and ethical predicaments ([167]).

Interventions targeting the collective-interior quadrant focus on shared societal beliefs, morals, values, and norms about pain. A sample of approaches includes public awareness campaigns and policy advocacy (e.g., [62]), pain education (e.g., [120]) and conventional group approaches (e.g., CBT, ACT ([247])) to help communities and individuals explore and reconceptualise pain. The body of knowledge on the culture, morals, and ethics of pain from sociology, cultural anthropology, and cultural psychology is diverse, though often overshadowed by biomedical research ([184]). Neglect of this body of evidence may have negative impacts, such as reinforcing racist, antiethnic, and sexist attitudes and beliefs within science and medicine, leading to the neglect and mistreatment of marginalised communities ([156]).

The healthcare sector, influenced by governments and corporate industries, dominates the narrative and culture of pain in WEIRD society, with religion also playing a role. The interconnected narratives and actions of several pharmaceutical companies that shape the intersubjective culture of pain in society have significantly influenced the ‘opioid crisis’ ([129]). Future directions include promoting the social model of pain to reduce stigma and inequalities and shifting societal mindsets from pathogenic to salutogenic through constructive pain language and health promotion strategies ([97]).

#### 3.1.4. Collective-Exterior: Systems and Structures

The collective-exterior quadrant reflects societal systems and structures, both physical and abstract, shared by communities at various levels. It is the ‘setting-centric’ aspect of the biopsychosocial model of pain, focusing on the places where daily activities occur and where environmental, organisational, and personal factors interact, such as schools, workplaces, and cities ([249]). Central to this quadrant is the socio-ecological habitat, where social and ecological systems interact, influencing pain through factors like economy, policies, industries, and environmental hazards ([7]; [159]).

Interventions targeting the collective-exterior quadrant focus on the shared systems and physical structures in the socio-ecological environment. A sample of approaches targeting the settings in which people live includes urban planning and design, policy and legislation, and environmental health interventions that aim to reduce inequality by improving the conditions in which people are born, grow, live, work, and age. This includes accessibility and quality of healthcare services for pain care by enhancing healthcare infrastructure, insurance coverage, and the availability of specialised and non-specialised pain services.

Research on the environmental and social determinants of pain from epidemiology, sociology, public health, health policy, systems theory and ecology is growing and gaining prominence, providing evidence of the influence of socio-economic inequality on pain outcomes and on the distribution of healthcare services and support (e.g., [7]; [47]; [110]; [178]; [227]). This is underpinned by research on physical and mental health (e.g., [188]; [225]).

The healthcare sector is the primary setting for pain management, and its policies, practices and culture is under the influence of systemic factors including the government, financial, business, media and technology sectors, and religion ([225]). Future directions for research include investigating macro-level geo-political forces that influence pain and associated issues ([92]; [127]; [255]) and developing health policies to create supportive environments (healthy settings), reduce inequalities, and promote healthy attitudes and behaviours towards pain ([104]; [187]).

Based on the author’s domain knowledge and the author’s interpretation of Wilber’s AQAL framework, specific features of pain and healthcare approaches mapped to each quadrant are summarised in Figure 6 and Figure 7, respectively.

This mapping to quadrants draws attention upstream and downstream influences on pain (Figure 8a). In the context of health, “upstream” refers to the root causes and structural factors of the shared environment that influence health, while “downstream” refers to interventions that focus on the treatment of illnesses and behaviour changes in individuals, i.e., once pain happens. Upstream and downstream domains can be divided further to illuminate healthcare approaches related to each quadrant (Figure 8b). To date, attention has focused on downstream interventions that deal with pain and its consequences after it has occurred in the individual. Focusing upstream to create healthier collective settings of the lower quadrants would address factors that exacerbate pain and prevent pain from resolving, i.e., that makes pain ‘sticky’ ([99]; [101]).

#### 3.1.5. Integral Model

The interconnectedness of collective culture (LL) and settings (LR) with contemporary neuroscience (UR—([29]; [115])) and contemporary psychology of pain experience (UL—([34])) is represented in Figure 9. The author crafted an embryonic integral narrative to explain pain as an emergent property from the interconnectedness of the four quadrant dimensions as follows: Changes in the internal environment of the body (UR) and external environment of the world (LR) are transduced into neural activity that may manifest as an embodied, embedded, emotive inner experience of pain (UL) and enacted behavioural output (UR) situated within an shared cultural intersubjective space (LL). An individual’s inner experience of pain manifests from neural networks operating on principles of active inference and generative predictive models that are refined according to ‘prediction errors’ resulting from the success or otherwise of the outcome of the output (inner experience of pain and behavioural action). The neural generative prediction model is optimised through bioplasticity (neural ‘tuning and pruning’) based on the fidelity of the prediction error (Prediction Fidelity) ([29]; [34]; [115]).

### 3.2. Mapping Pain to Levels

In this section, the author’s domain knowledge is mapped to Wilber’s levels of worldview and psychological development. The mapped data was reviewed, and a narrative was crafted, drawing on contemporary literature to illustrate and interpret the data in relation to the levels of development and study objectives.

#### 3.2.1. Evolving Worldview

The results of mapping aspects of pain to evolving worldview are provided in Table 3.

In this section, aspects of each level of worldview are presented in a narrative informed by contemporary literature.

In the Archaic pre-modern level, consciousness was likely to be formless, with no cultural manifestations or differentiation between humans and the world ([69]). Instinctual behaviour dominated, and pain motivated protective actions (e.g., guarding) and avoidance of harmful agents, with little meaning attributed to pain per se ([230]). It is probable that soothing methods available in the natural environment, like warmth or cold, would be used to alleviate unpleasantness, with minimal attribution of meaning.

In the Magic pre-modern level, consciousness was likely to be characterised by magical thinking, perceiving the world as animated and interconnected through mystical forces ([69]; [200]). Pain may have been attributed to supernatural causes, with coping mechanisms involving spiritual leaders and community rituals. An individual, believing that pain is a curse or bad karma, may have used ritualistic practices for relief.

In the Mythic pre-modern level, individuals developed structured worldviews based on myths and narratives, with identity tied to cultural stories and traditions ([69]; [116]; [154]). Pain may have been understood through religious or cultural stories, as a test of faith or a rite of passage, with coping mechanisms involving traditional healing practices and community support ([45]; [158]).

The modern level reflects rationalist methodologies rooted in Greek philosophy, that evolved significantly during the Enlightenment in the 17th and 18th centuries, emphasising individualism and objective analysis, leading to a medicalised WEIRD worldview of pain ([71]; [79]). In the modern rational level, individuals develop critical thinking and objective reasoning, valuing logic, empirical evidence, and scientific inquiry. The modern worldview prioritises individual rights, progress, and innovation, focusing on rational analysis and questioning traditional beliefs. Pain is understood scientifically, with emphasis on tissue, pathology, biological mechanisms, diagnosis, and a tendency towards biomedically orientated treatments ([71]; [135]). Coping mechanisms would include seeking specialists to undergo diagnostic tests and medical treatment plans based on scientific explanations and research. If unsuccessful in finding a precise tissue-based diagnosis and/or treatment, individual’s might engage in therapy shopping and experience an adversarial relationship with the healthcare system.

The post-modern pluralist level, emerging in the mid-20th century, is marked by pluralism and relativism, recognising diverse perspectives and the subjective nature of experiences and living reality. The post-modern, pluralistic level values multiple viewpoints and prioritises diversity, equality, and inclusion, engaging in critical discourse on cultural and social issues. Coninx et al. suggest that a pluralistic view is the only way to do justice about pain from a first-person perspective ([35]). The pluralistic worldview on illness emphasises social justice, environmental concerns, and contextual truth, focusing on community and shared values, while critiquing established structures ([137]). There is a flexible understanding of pain—as multifaceted and influenced by biopsychosocial factors, social inequalities, and social injustices ([25]). However, valuing various viewpoints risks erroneous thinking and irrational ideas regarding diagnosis and treatment and a “crisis of faith in pain medicine” that may be detrimental to long-term health and well-being ([83]; [139]). Coping mechanisms include biopsychosocial approaches, valuing medical and non-medical interventions, including psychological and complementary therapies from diverse healthcare practitioners and support groups.

The integral holistic level, emerging in the late 20th century, synthesises insights from previous levels, respecting diversity while recognising that some perspectives are more comprehensive or effective than others. The integral level unites knowledge from various levels to create a holistic understanding of reality, promoting collaboration across disciplines, cultures, and worldviews ([237], [238], [239], [242]). At the integral level, pain may be seen as an important part of life, offering opportunities for growth and self-understanding. Individuals develop a comprehensive understanding of pain by integrating insights from all previous levels. Coping mechanisms emphasise personal agency, mindfulness, holistic health strategies, and fostering connections with others. An individual with an integral perspective accepts pain as a catalyst for personal development, engages in spiritual practices, and advocates for others. They view pain as an opportunity for growth, improved relationships, and a deeper sense of being, employing diverse strategies to live a meaningful life informed by pain.

Transcendent experiences are described as ego-dissolving encounters with something greater than oneself and are cross-cultural, reported throughout history, and are the foundation of many religions ([6]; [72]; [114]). Wilber contends that the transcendent level focuses on a reduced sense of self and enhanced connection to a wider whole by incorporating spiritual, mystical, and transcendent experiences beyond ego, self, and space-time ([237], [238], [239], [242]). Wilber contends that this level integrates logical thinking, analysis, and empirical evidence with experiences of unity with nature and the universe, offering peace that surpasses simplistic rational thought.

At the transcendent level, it is likely that pain is seen as a state of consciousness that offers deeper meaning and purpose and is integrated into a holistic appreciation and understanding of a life journey as part of a profound connection to a greater whole ([68]). The transcendent individual would integrate intuition and analytical thinking and reason, accepting contradictions and paradoxes to see pain beyond the self and immediate habitat; pain would offer personal and subjective insights to *being* and *becoming* ([114]). Coping mechanisms attempt self-transcendence through spiritual practices, meditation, mindfulness, and building connections and social bonds with like-minded individuals and nature ([41]; [74]; [75]; [207]). The transcendent individual works towards a sense of peace with pain and explores their experiences beyond personal selfhood, viewing pain as a constructive part of a meaningful life journey and engaging in spiritual practices to explore being, belonging, and becoming within a broader transpersonal context. Reed and Haugan contend that self-transcendence is a salutogenic process to foster well-being ([171]).

#### 3.2.2. Psychological Development

The results of mapping pain to psychological development are provided in Table 4.

In this section, aspects of each level of psychological development are presented in a narrative informed by contemporary literature.

Wilber describes the pre-personal level as a primitive stage of psychological development in newborns, infants, and archaic adult hominids, where the individual lacks a sense of identity and cannot differentiate their experiences from ‘others’ ([239]). It is likely that pain would be experienced as a formless sensory and emotional context, with coping dependent on external support for comfort and soothing, eventually leading to self-soothing behaviours, as seen in infants and children ([105]; [119]).

The personal level involves the development of emotions and physical sensations as part of individual identity, including logical mental structures and abstract thinking (aligned with the modern individualist and post-modern pluralist worldview levels described previously). Pain would be experienced within ‘me’ and manifested as a personal ‘mosaic’ of experiences and influences ([61]) which may burden the sense of self and affect daily functioning ([254]). Coping strategies would be individualised, aiming to alleviate personal distress through treatments, therapies, and self-care, including conventional medical and psychological interventions.

The transpersonal level transcends individualism, fostering a broader awareness of interconnectedness with others and the environment (aligned with the transcendent worldview level described previously ([6]; [72]; [114]). Pain would be viewed as part of a deeper and broader human experience of exploration and growth incorporating deeper states of conscious and spiritual awareness. Coping strategies would include holistic and spiritual approaches through community support, and including traditional Eastern practices e.g., meditation, yoga, and faith-based practice in conjunction with conventional healthcare ([84]; [173]). Pain would be seen as an opportunity for curious exploration and personal growth, fostering empathy and compassion.

Wilber contends that individuals progress through sub-categories of transpersonal development in order to dissolve the ego and realise the oneness of all that exists ([238], [239], [242]). The scientific credibility of such sub-categorisation lacks empirical evidence and validation but may be a speculative ‘route map’ for transformation through pain and suffering in which individuals progress towards heightened intuition about emotional issues, non-duality, compassion, and acceptance. A mapping of pain to Wilber’s subcategories of transpersonal pain is provided in Appendix A.

#### 3.2.3. States of Consciousness

In the context of pain, states of consciousness refer to transient conditions of awareness that influence moment to moment pain experience, including sensory, affective, and cognitive dimensions of pain such as interpretation, meaning, and coping. States of consciousness include sleeping, awake, or altered states induced by medication or mindfulness practices. Recognising and exploring the impact of different states of consciousness on pain experience adds an additional layer of understanding and complexity.

The results of charting the distinguishing features of levels of development and states of consciousness for pain, based on the author’s interpretation of Wilber’s AQAL framework, are provided in Table 5.

### 3.3. Applying Levels to Healthcare

In this section, the author’s domain knowledge is used to craft a narrative to illustrate the application of mapping of modern, post-modern, and integral levels in the context of the pain patient, the healthcare practitioner, and healthcare services.

#### 3.3.1. The Pain Patient

This example illustrates how a pain patient’s attitudes and beliefs may vary according to their worldview.

A patient with a modern (individualist) worldview attributes their pain to a specific physical cause, such as tissue damage or a medical condition, believing that understanding and treating tissue alone will alleviate their pain. They see pain as a problem to be solved through scientific and medical means, focusing on objective diagnoses, e.g., using magnetic resonance imaging (MRI) and X-rays. They prefer scientifically proven treatments, such as medication or physical therapy, and trust their healthcare practitioner’s recommendations based on clinical data. This aligns with a mechanistic, biomedically orientated mindset of pain.

A patient with a post-modern (pluralist) worldview views their pain as a complex experience influenced by physical, psychological, and social factors, recognising that it might not have a single, clear cause and can be affected by stress, emotions, and social interactions. They seek a multi-perspective understanding of their pain and value treatments that address all aspects of their well-being, considering physical, emotional, and social factors. They might seek input from various healthcare providers, including psychologists, physical therapists, and alternative medicine practitioners, and appreciate a treatment plan that includes their firsthand experiences and preferences. This aligns with a biopsychosocial orientated mindset of pain.

A patient with an integral worldview views their pain as a multifaceted experience involving physical, psychological, social, and spiritual dimensions. They attribute meaning to their pain by considering its effects on various aspects of their life. They view pain as an opportunity for personal growth, improved relationships, and a deeper sense of being interconnected with the biosphere and the broader universe. They actively take part in advocacy for others with similar experiences, embodying the idea that pain is integral to a life well lived. They integrate a wide variety of strategies and techniques from diverse disciplines, including spiritual practices, to alleviate suffering and explore the complexity of their pain. This aligns with a holistic, whole-person approach to understanding pain and may lead to the development of transcendental practices and states.

#### 3.3.2. The Healthcare Practitioner

This example illustrates how a healthcare practitioner’s attitudes and beliefs may vary according to their worldview.

A healthcare practitioner with a modern (individualist) worldview focuses on objective, measurable data to diagnose and treat tissue believed to be causing the pain. They rely on medical tests (e.g., using MRI scans and X-rays) to identify any physical abnormalities. Treatment is based on evidence-based practices, such as prescribing medication or recommending physical therapy, with decisions driven by clinical data and scientific research. It is possible that they dismiss the value of psycho-social-spiritual interventions. This aligns with a mechanistic, biomedically orientated mindset, and is considered appropriate to *treat* acute pain, but less appropriate for long-term pain, e.g., chronic primary pain.

A healthcare practitioner with a post-modern (pluralist) worldview uses multimodal approaches and values being part of a multidisciplinary team that explores the influence of biological, psychological, and social factors on the patient’s pain experience. The healthcare practitioner seeks to create a comprehensive pain management plan tailored to the person’s individual needs informed by Western medicine and delivered by health care professionals (e.g., physicians, psychologists, and physical therapists), with support from complementary practitioners and community-based services. Although each perspective is considered equally important, priority is given to healthcare rather than non-healthcare solutions such as arts and crafts that are considered beyond scope. It is possible that they dismiss the value of non-conventional interventions e.g., transcendental and spiritual practices and alternative therapies. This aligns with a biopsychosocial multi-disciplinary healthcare-focused mindset and is considered appropriate for *managing* long-term pain.

A healthcare practitioner with an integral worldview unifies the strengths of both rational and pluralistic approaches and extends this using *inter*disciplinary input beyond healthcare-focused solutions. They use objective medical data and subjective experiential data of the individual, and objective and subjective data from the context of their collective communities in their diagnosis and strategies for support. An integral healthcare practitioner might combine conventional medical treatments with complementary therapies (e.g., mindfulness or acupuncture) and community-based support (e.g., yoga, arts, faith-based activities), and may advocate exploration of experiences of existence and the search for meaning in a life with pain through transcendental and spiritual practices. The goal would be to provide opportunities for the patient to reconceptualise pain and suffering in the context of a meaningful and purposeful life journey. This aligns with a spiritual-socio-psycho-bio transdisciplinary mindset where the patient is supported as a whole-person in a transformational journey of growth through pain (salutogenesis).

#### 3.3.3. Healthcare (Pain) Services

This example illustrates how attitudes and beliefs towards pain services may vary according to worldview.

A modern (individualist) worldview is reflected in hospital-based medically led services focusing on objective diagnosis of tissue pathology and the use of biomedical procedures and treatments to resolve pathology and/or relieve pain. This is appropriate for in-patient care of acute physical trauma and disease (e.g., in hospital care of post-operative pain).

A post-modern (pluralist) worldview is reflected in clinic-based therapy-led services of multidisciplinary healthcare practitioners using a biopsychosocial approach of integrated care based on personal pain management plans and a variety of medical and non-medical interventions delivered by healthcare specialists, e.g., physicians, psychologists, physical therapists, and possibly complimentary therapists with additional access support from the Voluntary, Community, and Social Enterprise (VCSE) sector. This reflects current recommendations for integrated pain service provision for chronic (long-term) primary and secondary pain by pain organisations e.g., the IASP ([88]).

An integral worldview would be reflected in culturally adapted pain support embedded in local community services that encompasses both modern and post-modern perspectives within its framework. Community-based pain support would utilise a collaborative patient–community–VCSE–healthcare venture that unifies diverse perspectives from Western and Eastern philosophies. Such a service would align requirements according to specific needs of the local community. Healthcare practitioners would triage patients. In instances where trauma or disease requires pathological repair, the patient would be referred to a clinically led hospital-based service. For long-term (chronic) pain without sinister pathology or mental health issues, patients would be referred to community-based support services. This view is consistent with UK government guidelines that recommend shifting services from hospital into community settings to support people with long-term conditions ([140], [141], [142]). The implementation of this care model in practical settings has encountered significant challenges ([60]).

The integral worldview has mutual respect for different service providers, approaches, and interventions (e.g., spiritual-socio-psycho-bio and Eastern and Western philosophies). Funding would be configured to ensure equity in power, including VCSE sector-led services supported by healthcare providers. This approach ensures services are situated and contextualised within local communities to foster holistic whole-person support tailored to the meaningful needs of the individual.

## 4. Discussion

The aim of this study was to explore the utility of a simplified version of Wilber’s AQAL framework to advance a biopsychosocial understanding of pain in a healthcare context. The simplified AQAL framework compares favourably to other meta-frameworks. The Dahlgren & Whitehead socio-ecological framework is widely adopted for mapping the relationship between individuals, their environment, and health determinants ([39]). Socio-ecological frameworks have been used to map biopsychosocial domains for chronic pain management and opioid tapering ([253]), understanding medical staff and patients’ views of chronic pain ([130]), and experiences of treatment in people who use opioids ([109]). Unlike the simplified AQAL framework, none of these socio-ecological frameworks account for subjective–objective dimensions or dynamic psychological changes over time.

Bronfenbrenner’s bioecological systems framework divides the environment into five components (microsystem, mesosystem, exosystem, macrosystem, and chronosystem) and has been used to explain how factors in these sub-systems are directed at the individual or society factors in the context of adolescent-centred pain management in school health care ([149]) and the treatment of paediatric pain ([123]). Unlike the simplified AQAL framework, Bronfenbrenner’s framework does not incorporate multiple dimensions and psychological levels of reality.

Numerous integrative frameworks have been applied to embed a biopsychosocial approach in the context of pain treatment and management ([113]; [117]; [204]) and service provision (e.g., the WHO’s Integrated People-Centred Health Services (IPCHS) Framework, the NHS England’s Health and Justice Framework). Unlike the simplified AQAL framework, they do not attempt to integrate first-person pain experience through subjective, objective, individual, and collective dimensions.

Traditional psychological frameworks tend to focus on specific aspects of human behaviour and mental processes, such as cognitive, behavioural, humanistic, psychodynamic, and biological theories, each addressing different facets of psychology. Integral psychology frameworks are more focused on integrating psychological theories and practices, without necessarily incorporating broader philosophical or spiritual dimensions. Frameworks to integrate mindbody (subjective–objective) do not incorporate individual/collective and interior/exterior dimensions or psychological development. For example, The Multimodal Assessment Model of Pain (MAP) attempts to integrate subjective and objective assessments of pain, emphasising the importance of patient narrative and compassion-based and mechanism-based management approaches ([234]). The MAP focuses on the nuanced assessment of pain in clinical settings rather than an integral understanding of the biopsychosocial nature of pain as a dynamic, subjective, evolving experience.

The main finding of mapping the quadrants was a comprehensive coverage and illumination of all dimensions of pain experience. This can assist conceptual coherence and a holistic understanding and highlights perspectives that are often neglected, i.e., of the lower quadrants. This opens opportunities for upstream solutions that address socio-ecological factors associated with culture and settings that contribute to chronification and treatment-resistance of pain ([96]). To date, attention has mostly been focused on downstream solutions of managing pain *after* it has started that often target the upper quadrants ([201]).

The main finding of mapping levels was to reveal a trajectory of conceptualisations and interpretations of pain in individuals and collectives, from modernist biomedical to post-modern biopsychosocial to integral holistic perspectives. This offers a route map towards an integral vision of pain that aligns with calls for a cultural transformation of healthcare towards whole-person community focussed support ([24]; [67]).

The following sections discuss the implications of these findings to advance a biopsychosocial understanding of pain (objective 1), promote holistic healthcare practice (objective 2), and guide future directions in healthcare services and research (objective 3).

### 4.1. Advancing an Understanding of Biopsychosocial Pain (Objective 1)

The AQAL framework extends the biopsychosocial model without challenging its foundational principles. The biopsychosocial model focuses on the individual, whereas the AQAL framework integrates individual and collective, and interior (subjective) and exterior (objective) dimensions and psychological development. The quadrants assist in organising pain theory focusing on various viewpoints, e.g., neuroscience (e.g., ([115])—UR), psychology (([34]; [199])—UL), and settings (([104])—LR). This enables development of an integral model of pain that captures all quadrants to advance conceptual coherence in the context of an evolving understanding of the meaning and purpose of pain in the milieu of modern living. In this regard, the AQAL framework complements the philosophy of Unitary Caring Science, valuing human experiences through an evolved holistic and humanising lens, fostering a deeper understanding of the interconnectedness of mind, body, and spirit ([231]; [232])

Using the quadrants framework illuminates a conceptual error in pain science—the reification of pain ([32]). In philosophical and psychological contexts, reification involves attributing physical characteristics or object-like qualities to something that is inherently subjective and experiential, like pain. The visual nature of the quadrant framework adeptly exposes this misconception because pain locates in the upper left not upper right quadrant. Likewise, quadrants illuminate the error of conflating pain and nociception ([4]; [32], [31]). Thus, the quadrants advance conceptual coherence of the biopsychosocial model.

The quadrants framework has potential to guide the analysis of existing theories. For example, the ‘5E’ process of pain ([34]; [199]) could be explored by mapping features and influences of pain experience (UL) to quadrants:Embodied: Pain experienced in body parts (UR).Embedded: Pain influenced by environment and context (LL/LR).Enacted: Pain shaped by actions and behaviours (UR).Emotive: Pain driving action (UR).Extended: Pain extending to interactions with the world (UR/LL/LR).

### 4.2. Promoting Holistic Healthcare Practice (Objective 2)

Major criticisms of the biopsychosocial model include that it is fragmentated and decontextualised from living experience. The quadrants framework has potential to interlock dimensions improving conceptual coherence for pain education, empowering layperson narratives. For example, “*My inner experience of pain (UL) is real and emerges from tissue in my body (UR) influenced by my (our) culture (LL) and my (our) social settings (LR)*”. The quadrants framework is amenable to everyday language making pain education understandable, approachable, and meaningful, and avoiding negative connotations associated with terms like ‘psychological’ which may imply ‘pain is in the head and not ‘real’ ([16]). This makes conversations about pain empowering and less stigmatising, and supports constructive, person-centred clinical conversations about the nature, reason and meaning of pain that foster empathetic care and a shared decision making approach ([176]; [183]; [229]).

Mapping illuminates the individualistic nature of conventional clinical biomedical and psychological approaches of the upper quadrants (e.g., surgery, drug medication, neuromodulation therapies, behaviour modification, CBT, ACT). Quadrant mapping brings the shared contexts of culture and settings to the fore opening integral approaches. For example, supporting a person’s inner world using group-based approaches in non-healthcare cultures and non-clinical settings. such as artist-led activities delivered in community contexts ([58]; [100]). An example of such an approach is Unmasking Pain, an innovative artist-led project to explore creative approaches for exploring life with long-term pain that provided a ‘space to breathe’, ‘flexibility to participate’ and ‘the possibility for change’ ([211]). Opportunities also arise for upstream solutions that mitigate socio-ecological factors contributing to chronification and treatment-resistance of pain, i.e., that make pain ‘sticky’ ([96]; [17]).

An important finding to emerge from mapping pain to levels was to highlight the evolution of conceptualisations and interpretations of pain from a modernist biomedical to a post-modern biopsychosocial perspective. Modernity socialises people that pain is resolved by identifying and treating physical causes through scientific means ([8]). Recent evidence suggest that societal beliefs about pain may be shifting towards contemporary, multidimensional understandings of pain in line with a post-modern biopsychosocial worldview ([168]). An integral vision for pain advances the current fragmented decontextualised biopsychosocial view by encompassing interconnectedness to the collective culture and settings of society. These aspects of pain experience are upstream and under the influence of macro-level societal forces and as a consequence are often considered beyond the remit of conventional healthcare practice ([44]; [190]).

Realising an integral vision of a whole-person-centred approach requires cultural transformation that recognises the limitations of the pathogenic biomedical model as a sole means of supporting people with long-term pain ([67]). The WHO advocate a social model of health to reduce stigma and foster healthier societal attitudes and supportive environments in which people flourish ([110]; [153]; [249], [252]). AQAL mapping positions pain within a social context that addresses the influence of evolutionary-mismatch and WEIRD bias on the chronification and treatment resistance of pain. An integral vision of pain would encompass cultural transformation from pathogenesis to salutogenesis to foster a whole-person, healthy-settings perspective of pain that is grounded in a social model of health and a person’s ‘spirituality’.

#### 4.2.1. Salutogenesis

Salutogenesis, a term coined by Aaron Antonovsky, emphasises the origins and resources that support health and well-being ([122]; [133]). Central to salutogenesis is a sense of coherence (pain is logical), grounded in comprehensibility (understanding pain), manageability (resources to cope with pain), and meaningfulness (the purpose of pain as a life experience) and the use of general resistance resources like personal traits, social support, and material resources to foster positive health behaviours and environments ([13]; [133]; [203]). The application of salutogenic principles in relation to healthcare support for pain is still in its infancy ([12]; [133]; [152]).

A salutogenic approach cultivates physical and mental growth consistent with evidence that people living with long-term pain consider healing to be an “ongoing and iterating journey rather than a destination” ([213]). Salutogenic approaches are holistic, person-centred, and compassionately validate pain though listening to and valuing life-stories, encouraging self-kindness, facilitating safe social reconnection, and exploring new possibilities ([13]; [133]; [213]). Salutogenic approaches focus on the growth of healthy minds and behaviours through meaningful understandings of pain via education, preventive programmes, environmental changes, economic incentives, policy making, social support networks, and community engagement to ensure cultural relevance. This is the domain of health promotion with opportunities for health promoters to influence the agenda to improve public health pain policy and practice ([99]; [101]; [251], [252]).

#### 4.2.2. Painogenicity and Healthy Settings

Evolutionary-mismatch may foster a painogenic environment defined as “the sum of influences that the surroundings, opportunities, or conditions of life have on promoting long-term pain” ([95]). There has been extraordinarily little scholarship on pain and evolutionary-mismatch—the idea that modern lifestyles of humans are maladapted for their genetic heritage, which is adapted to Paleolithic environments. Mismatch between Paleolithic physiology and modern lifestyle is likely to promote the persistence of pain ([244], [245]). Johnson and Woodall’s socio-ecological settings model of the painogenic environment illustrates how socio-ecological factors in WEIRD ableist societies may constrain healthy lifestyles and restrict recovery opportunities for people living with pain ([104]), i.e., make pain ‘sticky’ ([17]). Damage-loaded warmongering pain language may by an insidious force contributing to a painogenic environment suggesting a need for more constructive narrative ([14]; [95]; [102]; [104]; [144]). The AQAL framework is one means of raising public awareness about the painogenic environment and this may help to shift attention to health-promoting strategies to mitigate upstream factors in much the same way as the obesogenic environment has done for the challenge of obesity ([53]; [96]; [223]).

#### 4.2.3. Spirituality

AQAL mapping reveals an integral vision of psychological growth through interconnectedness and ‘spirituality’. Spirituality may be misinterpreted as supernatural or mystical, creating a false dichotomy between ‘scientific/rational’ and ‘pseudoscience/irrational’. Spirituality is increasingly being understood in the realm of healthcare in a broad, non-religious context, referring to personal growth. There is no consensus on a specific definition of spirituality, although a commonly accepted definition describes spirituality as an aspect of humanity that involves how individuals seek and express meaning and purpose and how they experience their connectedness to the present moment, themselves, others, nature, and the significant or sacred ([166]). Spiritual care is at the core of end-of-life care, especially in relation to cancer ([165]) and a cornerstone of ‘Total Pain’ championed by Cicely Saunders in the context of the hospice ([248]). Integral pain endorses spirituality as a core element of pain support, which is notably absent from current guidelines for non-cancer pain.

### 4.3. Future Directions in Healthcare Services (Objective 3)

#### 4.3.1. From Integrated to Integral Care

AQAL mapping advances contemporary healthcare viewpoints. Presently, pain organisations advocate *integrated/integrative* pain care, using biopsychosocial approaches delivered by multidisciplinary teams creating individualised care plans. Integrative pain care, as defined by the IASP, is a coordinated, evidence-based approach that combines conventional and traditional/complementary methods within a biopsychosocial framework, optimally delivered through an agreed model of care embedded in healthcare settings ([15]). The aspiration is to coordinate multiple healthcare providers to optimise comprehensive and continuous care, emphasising efficiency, quality, and patient-centredness that respects diverse healthcare practices and interventions without necessarily unifying them.

The aspirations of this integrative approach are proving difficult to implement and have yet to be fully realised, so the IASP annual outreach campaign for 2023 was ‘Integrated Pain Care’ and emphasised the need for non-drug, self-management care ([90]). The implication is that integrative care is an extension of existing healthcare services. There are at least two shortcomings of the integrated mindset and approach.

Firstly, the IASP define the integrative approach as “…the carefully planned integration of multiple evidence-based treatments—offered to an individual suffering from pain—that strives to be individualised (person-centred), mechanism-guided, and temporally coordinated.” ([70]). Often integrated pain services are developed using existing health service frameworks that value mechanistic mindsets that prioritise simplistic, single treatment interventions of the upper right quadrants that are amenable to randomised controlled clinical trial evidence. This is at the expense of more complex and messy interventions needed to tackle the socio-ecological influences of the collective of the lower quadrants.

Secondly, the complexity and cost–benefit–risk profiles of combining multiple healthcare services, treatments, and clinicians results in higher costs, increased utilisation of healthcare personnel, and more intricate logistical planning ([9]; [212]). There can be a lack of shared understanding and motivation to collaborate among clinicians, known as the “silo effect”, and dissonance between clinicians and patients ([87]; [118]; [170]).

Gaudet argues that the greatest threat to improving population health and well-being is failing to recognise that ‘true’ transformation, not just improvement, is needed across multiple systems ([67]).

#### 4.3.2. Integral Transformation of Services

Healthcare policy emphasises the importance of empowering individuals in their care yet first-person research demonstrates adversarial relationships between patients and healthcare providers, due, at least in part to misalignment in values and expectations ([46]; [214], [215], [216]). A meta-ethnography of 195 qualitative studies ([213]) found that people see long-term pain as an ongoing journey rather than a destination and health interventions would better focus on the following:Validating pain through meaningful explanationsValidating patients by listening to their storiesEncouraging patients to connect with a meaningful sense of self and explore new possibilitiesFacilitating safe reconnection with the social world.

Integrative pain services, with multiple mechanism-guided biological and psychological treatments delivered within clinical frameworks, are not designed to address the central tenets of experiential healing. Their focus is much more on downstream factors. Gaudet argues that to improve healthcare system outcomes the focus must shift to discovering what gives individuals meaning and purpose in life ([67]). Medical and therapy-led services may not be the best service provider to achieve this.

An integral service refocuses pain upstream with potential for ‘true’ cultural transformation through salutogenesis and healthy settings adapted for local communities. For this to happen, equal partnership between community members, the VCSE, and healthcare providers is required to co-produce culturally relevant pain support.

Such system change in healthcare service provision has been advocated by UK health policy for over a decade ([192]; [193]; [194]; [197], [198]) but has been difficult to implement in practice, with services inevitably drifting to mechanistic upper quadrant interventions ([185]; [209]; [212]). Typically, public sector funds and, therefore, ‘power’ are for healthcare-led pain services, sidelining patients and the VCSE sector organisations to a secondary role. An integral vision of pain advocates community-empowered pain support to meet local needs through strong partnerships with the VCSE sector.

#### 4.3.3. Bringing an Integral Vision to Life: The Rethinking Pain Service

Advocacy for community services for pain has been longstanding but rarely realised. To truly transform culture and care, a shift in financial power is essential. In 2022, an Integrated Care Board in England took a courageous decision to redistribute public healthcare funds to a VCSE sector organisation called [112] ([112]), to deliver a community-based pain support service called [172] ([172]). Although not designed using an integral framework, the Rethinking Pain service aligns with the principles of an integral vision of pain and demonstrates that cultural transformation is possible when public funding is reconfigured from clinic to community.

Rethinking Pain serves a culturally diverse and socioeconomically deprived population in Bradford District and Craven, and is the first public sector funded VCSE-led pain service in England (i.e., commission by the Integrated Care Board and overseen by the NHS). Rethinking Pain integrates the VCSE and clinical sectors to provide holistic, community-based support for long-term pain and was co-produced with the community to offer personalised, culturally relevant care.

The Rethinking Pain team, with health coaches at the core supported by clinical colleagues, engages with the community, particularly those facing health inequalities, offering a holistic care pathway. The Rethinking Pain service offers a comprehensive three-tier service user (patient) pathway for adults with long-term pain, helping individuals set goals and develop coping strategies through personalised care plans, one-to-one health coaching, community-based activities, and various pain education modules.

Tier 1 involves a two-hour interactive “Understanding Pain” workshop available in community settings and multiple languages. Tier 2 provides ongoing adult education, including workshops on “More on Managing Pain”, “Keeping Active and Safe Movement”, “Sleep Therapy”, “Diet Therapy”, “Emotional Well-being Support”, “Developing Helpful Habits and Setting Goals”, “Creative Therapies”, “Your Story”, and “Faith, Beliefs, and Pain.” Tier 3 directs more complex people to psychological therapies such as CBT and ACT.

The Rethinking Pain model not only alleviates pressure on clinical services but also promotes social connectedness and empowerment. By tailoring care to socio-cultural contexts, it addresses health inequalities and leads to sustainable health interventions. For a comprehensive description of the Rethinking Pain service and its effectiveness in reducing the burden on clinical services, as well as feedback from service users, please refer to the review by [103] ([103]).

### 4.4. Limitations

Wilber’s integral theory has been criticised for a lack of empirical validation through conventional peer review of academic journal articles ([22]; [42]; [76]; [136]; [150]; [161]). This could account for the limited exposure of Wilber’s scholarship in pain academe. For this reason, a simplified version of the AQAL framework was used in this study to avoid excessive analysis, overinterpretation, overextrapolation, and wayward discourse. The risk of wayward interpretation is likely to be greater for levels of development than quadrants. The granularity of and boundaries between levels are a matter of conjecture, and Wilber claims that levels of development are akin to spiralling streams and waves rather than purely linear ([240]). In the context of this study, the trajectory of psychological development is more important than boundaries between and within levels, especially at the transcendent and transpersonal levels that have not gained widespread acceptance within mainstream psychology ([64]). Until empirical validation is forthcoming the minutiae of Wilber’s AQAL framework, including transpersonal subcategories, should be considered speculative.

The importance of academic peer review is undisputed, but it does not guarantee conceptual rigor ([106]; [108]; [174]; [210]). There is substantial research on the application of the biopsychosocial model to pain and rehabilitation, including validation of tools for clinical practice ([91]; [177]; [219]), yet this has not prevented wayward discourse ([175]). Ensuring frameworks deliver valid, stable insights into healthcare research remains a challenge.

The subjectivity of the author was integral to the methodological process. Advocating for a particular position (i.e., holistic, person-centred healthcare) and reliance on the author’s domain knowledge can introduce cognitive biases such as selective reporting, confirmation bias, anchoring bias, availability bias, and reduced objectivity, which compromise the integrity and credibility of findings and distort the representation of information ([59]; [143]; [191]). Biopsychosocial pain literature predominantly focuses on the individual dimension of the upper quadrants, so the AQAL framework can mitigate cognitive bias by forcing engagement with the collective dimensions of lower quadrants found in the literature from the humanities, arts, and social sciences, including economics, geography, and ecology. However, the author is a non-clinical pain scientist and more conversant with subject disciplines of the upper rather than lower quadrants. The author attempted to mitigate the impact of cognitive biases through reflexivity, self-awareness, and embracing interdisciplinary perspectives and critical thinking, documented as essential reflections. The author also triangulated domain knowledge with published literature and debriefings with a network of colleagues (pain scholars) to safeguard a more balanced and informed approach.

As a post-positivist critical realist, the author reflected on balancing domain knowledge with openness to new ideas and reflection on selection bias. Debiasing mechanisms, such as cognitive forcing tools that help to counter biases in science were not used in this instance as they can be prescriptive and homogenise findings when judging literature across diverse approaches and multiple interpretative practices ([26]; [94]). This can constrain nonconformist perspectives and hinder capturing the diversity of personal pain experiences, where meaning and knowledge are understood as situated and contextual ([18]; [26]; [182]).

Despite these shortcomings, the quadrants are straightforward; they are ways of investigating phenomenon from different viewpoints, i.e., individual, collective, subjective, and objective. Conventional psychology accepts the notion of levels (stages) of cognitive development, including those chosen for this article, i.e., worldview (e.g., [86]; [116]; [132]; [208]) and psychological growth (e.g., [11]; [155]). Even if divisions between psychological levels are disputed, they still offer utility as a metaphorical roadmap to assist a deeper understanding of pain as an evolving experience within the socio-ecological milieu of modern living.

Finally, in this analysis the starting point was to conceptualise pain as a single undifferentiated entity rather than the complex, dynamic, often idiosyncratic, entity that it is ([37]). This may result in inferences that are prone to overgeneralisation, so caution is needed to translate rather than generalise the findings. Nevertheless, the quadrants were particularly useful in unpacking specific features of pain and may be valuable as a framework to explore more nuanced aspects of pain experience.

### 4.5. Future Research Directions (Objective 3)

Future research should validate the findings of this study with stakeholders, including individuals with lived experience of pain, scholars from various disciplines, healthcare professionals, and policymakers. Additionally, it should involve collaboration with domain knowledge experts from each quadrant and empirical testing and bibliometric analysis to identify influential papers within the cited literature, thereby mitigating selection bias.

The AQAL framework offers promising avenues to formulate more nuanced questions, objectives, and constructs for future exploration. Examples, include using the AQAL framework to explore lower quadrant influences on the persistence of different types of pain (e.g., ICD-11 categories, in-patient-outpatient contexts) and response to treatment or care. An interesting avenue of inquiry could be to analyse the influence of lower quadrant factors on treatment outcome. For example, a more nuanced understanding of the efficacy of a treatment intervention like acupuncture could be gained by analysing the interaction of an acupuncture needle with tissue (UR), the individual’s expectation of outcome (UL), the acupuncturist’s interaction with the individual (LL), the philosophical paradigm and treatment narrative (LL), and the treatment setting (LR). This would provide an integral understanding of acupuncture treatment.

Macrosociological research that utilises the AQAL framework may be useful to explore the influences of large-scale social structures and institutions such as economic and political systems and cultural norms contributing to broad sociological patterns and trends in pain. This would include mixed-method approaches and interviews, case studies, and ethnographic research to capture detailed personal narratives alongside statistical data, providing a comprehensive understanding of how macro-level forces impact individual experiences. Moreover, the utility, validity, and reliability of AQAL-informed policy adjustments, system changes, care plans, assessment tools, or educational programmes, especially those targeting population health outcomes would need to be evaluated against clear objectives using mixed-methods approaches ([186]; [212]).

Mapping AQAL to pain emphasises the importance of integrating and valuing broader research paradigms to deepen the understanding of pain and evaluate treatment outcomes. This includes positivist (quantitative ‘third-person’ methods [160]), interpretivist (qualitative ‘first-person’ methods [124]), critical (addressing power dynamics [2]), and pragmatic (mixed-methods [181]) paradigms, with methodologies tailored to specific evaluation goals, such as assessing effectiveness against outcomes beyond pain, like well-being and quality of life. Capturing feedback from all stakeholders, including individuals with pain, is crucial for refining interventions to remain effective and responsive to evolving needs.

## 5. Conclusions

The findings of this study are consistent with the hypothesis that applying a simplified version of the AQAL framework to pain provides insights that advance a biopsychosocial understanding. Mapping the quadrants ensures no fundamental aspects of pain experience are neglected, aiding conceptual coherence and holistic understanding. Mapping levels of psychological development highlights the evolution and variability of beliefs, attitudes, and understandings of, and values towards pain in individuals and collectives that influences the variability in experience and response to treatment. Mapping pain in this way advances conceptual coherence of the biopsychosocial model, drawing attention to upstream influences of culture and settings that may contribute to a painogenic environment, opening opportunities to reconfigure pain within a social model of health.

In conclusion, a simplified version of the AQAL framework effectively served as a heuristic device to explore subjective, objective, individual, and collective dimensions of pain, as well as psychological growth. This approach facilitated the development of an integral vision of pain, with potential to transform a fragmented, decontextualised biopsychosocial model steeped in pathogenesis into a coherent salutogenic humanistic paradigm. This integral paradigm empowers communities and VCSE sector services to deliver health promotion solutions, supported by healthcare providers, as exemplified by the UK’s Rethinking Pain Service.

## Figures and Tables

**Figure 1 behavsci-15-00703-f001:**
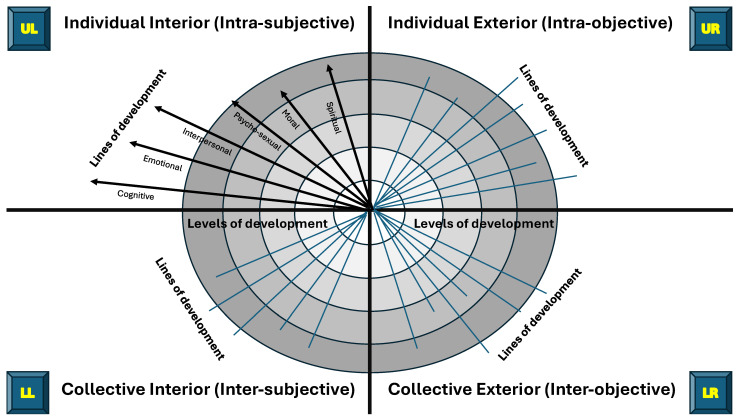
The author’s interpretation of Wilber’s AQAL framework primarily based on ([239], [242]). Quadrants are dimensions of individual-interior (psychological; upper left (UL)), individual-exterior (body and behaviour; upper right (UR)), collective-interior (cultural; lower left (LL)), and collective-exterior (systems and structures; lower right (LR)). Levels (stages) of human psychological development are represented as shaded circles ‘evolving’ outward from the centre of the grid.

**Figure 2 behavsci-15-00703-f002:**
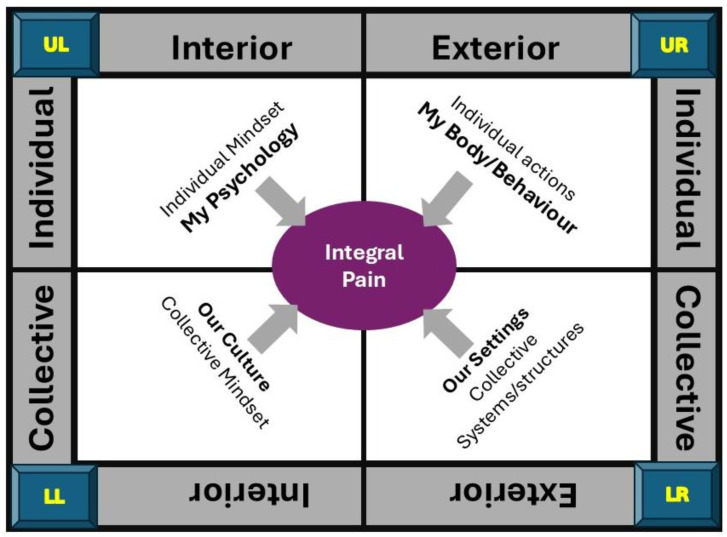
The author’s interpretation of Wilber’s quadrants primarily based on ([239], [242]).

**Figure 3 behavsci-15-00703-f003:**
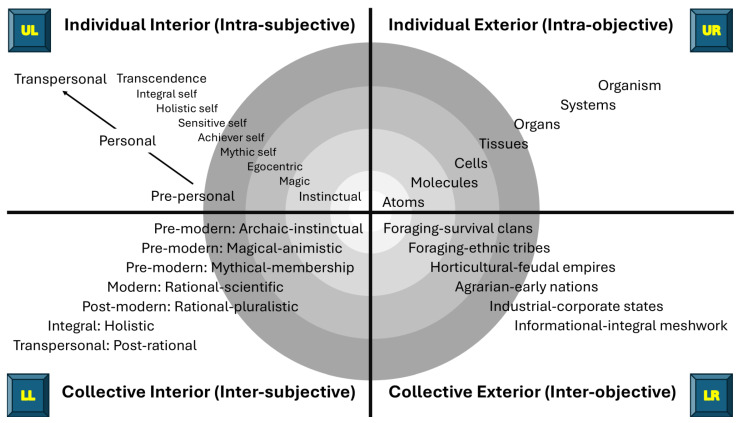
The author’s interpretation of examples of Wilber’s lines and levels of development for each quadrant primarily based on ([239], [242]). Lines of development: UL = cognitive self; UR = body structure; LL = worldview; LR = social structure and systems.

**Figure 4 behavsci-15-00703-f004:**
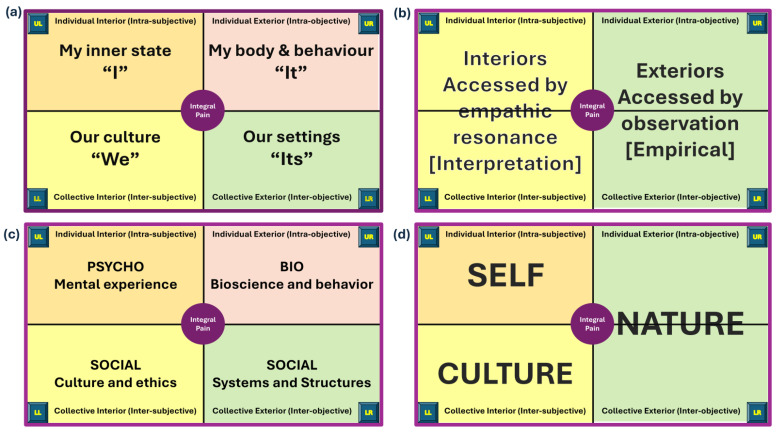
Quadrant perspectives. (**a**) Sobriquets (nicknames), (**b**) inner (subjective) and external (objective) domains, (**c**) biopsychosocial domains, (**d**) inner intra- and intersubjective minds and the external physical world (nature). Based on the author’s interpretation of Wilber’s AQAL framework ([239], [242]).

**Figure 5 behavsci-15-00703-f005:**
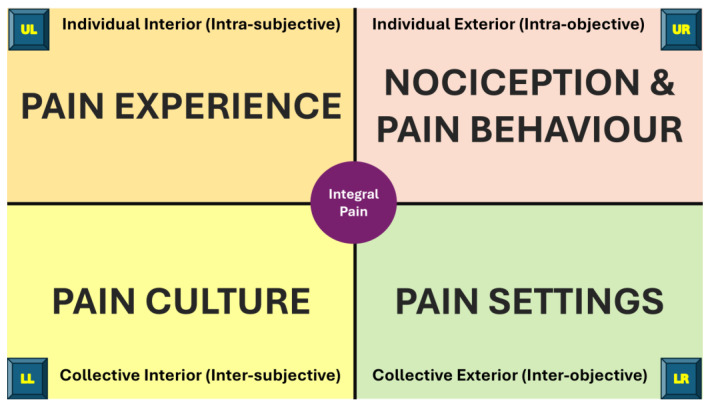
Quadrant perspectives for pain. Based on the author’s domain knowledge and the author’s interpretation of Wilber’s AQAL framework ([239], [242]).

**Figure 6 behavsci-15-00703-f006:**
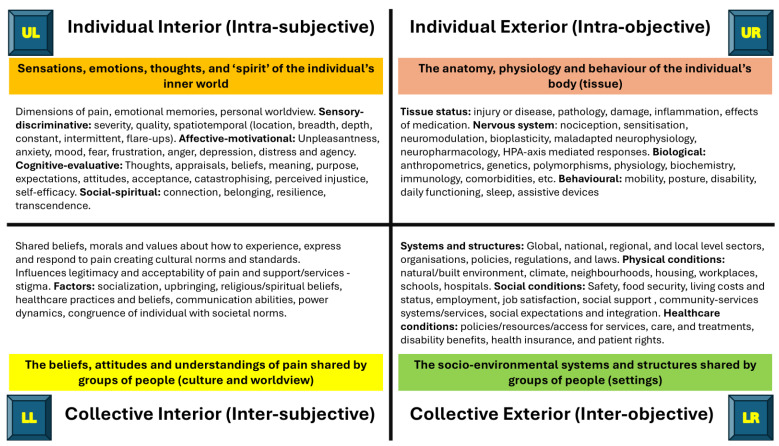
Summary of features of pain mapped to each quadrant. Based on the author’s domain knowledge and the author’s interpretation of Wilber’s AQAL framework ([239], [242]).

**Figure 7 behavsci-15-00703-f007:**
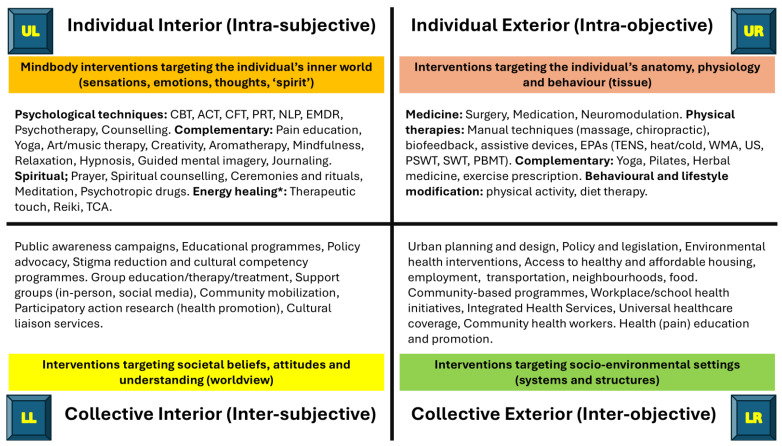
Summary of healthcare approaches targeting each quadrant based on the author’s domain knowledge and the author’s interpretation of Wilber’s AQAL framework ([239], [242]). * There is scepticism about the scientific trustworthiness of explanatory models for energy healing techniques. Key: TCA, traditional Chinese acupuncture—targeting meridians ‘energy flow’; WMA, Western medical acupuncture—targeting tissue; CBT, cognitive-behavioural therapy; ACT, acceptance and commitment therapy; CFT, compassion-focused therapy; PRT, pain reprocessing therapy; EMDR, eye movement desensitisation reprocessing; EPAs, electrophysical agents; TENS, transcutaneous electrical nerve stimulation; US, ultrasound therapy; PSWT, pulsed shortwave therapy; SWT, shockwave therapy PBMT, photobiomodulation therapy (laser).

**Figure 8 behavsci-15-00703-f008:**
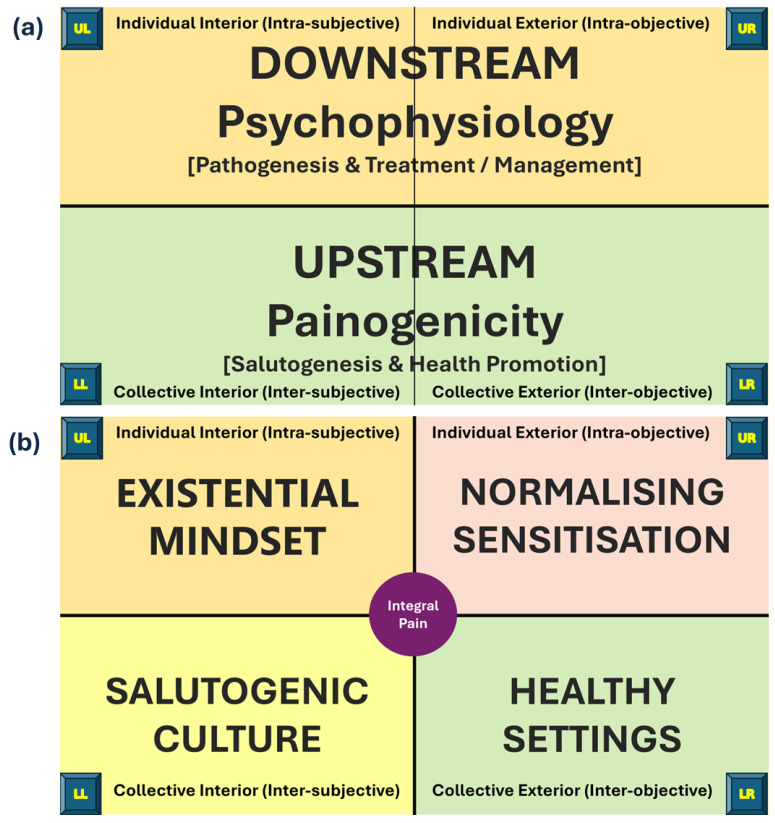
(**a**) Upstream and downstream mapping. (**b**) Quadrant solutions. Based on the author’s domain knowledge and the author’s interpretation of Wilber’s AQAL framework ([239], [242]).

**Figure 9 behavsci-15-00703-f009:**
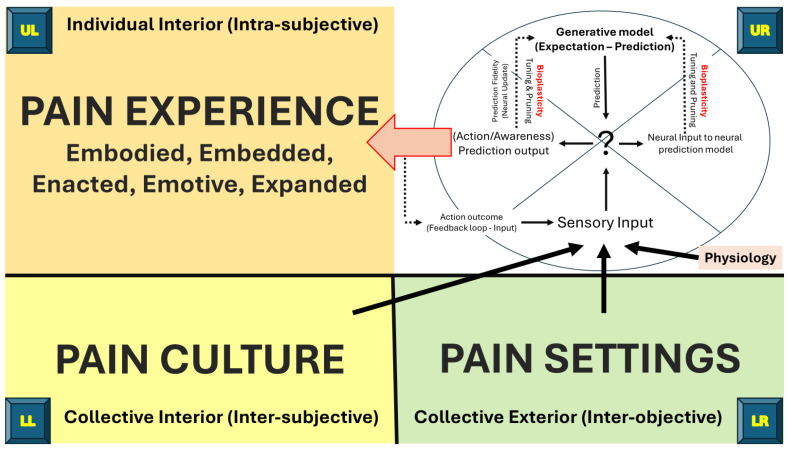
The interconnectedness of quadrants to inform an integral model of pain. Based on the author’s domain knowledge and the author’s interpretation of Wilber’s AQAL framework ([239], [242]).

**Table 1 behavsci-15-00703-t001:** The author’s interpretation of features of Wilber’s quadrants ([239], [242]).

Individual-Interior (Intrasubjective)	Individual-Exterior (Intraobjective)
**An individual’s inner state**	**An individual’s bodily actions (physiology and behaviour)**
**Perspective:** First Person (Subjective)—‘I’	**Perspective:** Third Person (Objective)—‘It’
**Focus:** Inner state of personal and introspective aspects of experience, emphasising individual psychological and spiritual development. Subjective experiences, thoughts, emotions, and beliefs, consciousness, intentionality	**Focus:** Objective and measurable aspects of the structure and function of the physical body. Structure, form, and functions of tissue, physiological responses, and observable behaviours
**Disciplines:** Psychology, phenomenology, and introspective studies (arts and humanities/qualitative)	**Disciplines:** Biomedical sciences, physiology, behaviour, biology (sciences/quantitative)
**Components:**	**Components:**
**Sensory, emotional, and cognitive domains**: Personal sensations, feelings, emotions, beliefs, ideas, and thoughts	**Body tissue/morphology:** Biological material
**Self-identity**: Personal sense of self and individual identity	**Body physiology**: Biological and physiological processes
**Sense of cohesion:** Personal sense that the world is comprehensible, manageable, and meaningful	**Behaviour**: Observable actions and behaviours of an individual
**Conscious development:** Personal awareness and subjective experiences	**Physical ability:** The limits of a person’s physical functioning
**Philosophical paradigm:** Interpretative, hermeneutic, phenomenology, and introspective psychology, emphasising the importance of personal consciousness and self-awareness	**Philosophical paradigm:** Empirical science, positivism, monological, focusing on measurable and observable biological phenomena
**Research focus:** Phenomenological studies, introspective methods, qualitative interviews, and self-report surveys to study subjective experiences, thoughts, emotions, and beliefs	**Research focus:** Empirical research, observations, scientific methods, experiments, measurements (physics, chemistry, biology) to study observable from and function, physiological processes and human behaviour
**Collective-Interior** **(Intersubjective)**	**Collective-Exterior** **(Interobjective)**
**Shared interaction of the inner states of people**	**Shared interaction with external world (systems and structures)**
**Perspective:** Second Person (Intersubjective)—‘We’	**Perspective:** Third Person (Interobjective)—‘Its’
**Focus:** Communal and cultural aspects of human experience, emphasising how individuals relate to each other within a shared context	**Focus:** Objective and measurable aspects of collective existence
**Disciplines:** Cultural anthropology, sociology, and cultural psychology (arts and humanities/quantitative)	**Disciplines:** Sociology, systems theory, and ecology (arts and humanities/quantitative)
**Components:**	**Components:**
**Ethics and Morals**: Shared values and principles that guide behaviour.	**Systems and Structures**: The organisational and systemic aspects of society, such as institutions, governments, and economies
**Worldviews**: Common perspectives and cultural narratives.	**Ecosystems**: Environmental and ecological systems
**Intersubjective Meaning**: Mutual understanding and shared experiences.	**Social Systems:** Patterns and structures within societies, including cultural norms and social institutions
**Culture:** The collective customs, arts, social institutions, and achievements of a particular group	
**Philosophical paradigm:** Interpretivism, constructivism, hermeneutics, and cultural anthropology, emphasising the importance of collective consciousness and social context	**Philosophical paradigm:** Systems theory, structural functionalism, and sociology, focusing on the external, structural influences on collective behaviour
**Research focus:** Ethnography, cultural analysis, discourse analysis, and participatory action research to study shared values, cultural norms, and collective worldviews	**Research focus**: Systems theory and analysis, social network analysis, quantitative surveys, and ecological studies to study social structures, institutions, and environmental factors

**Table 2 behavsci-15-00703-t002:** Mapping aspects of pain to quadrants. Based on the author’s domain knowledge and the author’s interpretation of Wilber’s AQAL framework ([239], [242]).

Individual-Interior (Intrasubjective)	Individual-Exterior (Intraobjective)
**A person’s experience of pain**	**A person’s bodily functions (physiology and behaviour)**
**Perspective:** An individual’s inner experience of pain—i.e., pain itself	**Perspective:** An individual’s physical body functions and behaviours associated with pain
**Focus:** Inner personal and introspective aspects of pain experience (Mind) e.g., How does pain ‘feel’ and what thoughts arise from pain?	**Focus:** Observable aspects of the physical body when pain is experienced (Tissue) e.g., How does pain emerge from physiology and impact on behaviour?
**Dominant disciplines:** Psychology, phenomenology, and introspective studies (arts and humanities/qualitative)	**Dominant disciplines:** Physiology, behaviour, biology (sciences/quantitative)
**Components:**	**Components:**
**Somatosensations**: Features of pain, such as location, quality, severity, spatial and temporal character, evoked response, allodynia, hyperalgesia	**Physiology:** nociception, sensitisation, bioplasticity, neuromodulation, neurotransmission, inflammation
**Emotions**: Feeling of, unpleasantness, distress, fear, depression, agency (need-state)	**Brain Function**: Neurophysiological correlates of pain
**Thoughts**: Beliefs and strategies about cause, impact, action, coping, wellness, meaning, prognosis	**Behaviour**: Escape, guarding, rubbing, avoiding, coping strategies
**Self-Identity**: Changes in sense of self and individual identity	**Physical Skills and Abilities:** Functional disability associated with pain
**Practitioners:** Psychologists, psychotherapists, psychiatrists, nurses	**Practitioners:** Physicians, physiotherapists, nurses, dietitians, occupational therapists
**Interventions (treatment/therapeutic):** Targeting feelings, thoughts, and inner experiences	**Interventions (treatment/therapeutic):** Targeting tissue and behaviour
**Collective-Interior** **(Intersubjective)**	**Collective-Exterior** **(Interobjective)**
**The societal view of pain**	**The social and physical environment**
**Perspective:** Shared beliefs about pain	**Perspective:** Shared systems and structures in the external environment that influence pain
**Focus:** The interconnectedness of shared communal and cultural values, perspectives, meanings, explanations, and understandings of pain, e.g., How does the worldview of pain influence pain experience?	**Focus:** Settings of collective existence (natural, built, and abstract) and their interconnectedness with pain experience, e.g., How do socio-ecological settings influence pain experience?
**Disciplines:** Cultural anthropology, sociology, and cultural psychology (arts and humanities/quantitative)	**Disciplines:** Sociology, systems theory, economics, and ecology (social sciences/quantitative)
**Components:**	**Components:**
**Culture:** Collective customs to endure, to seek help from physicians, healthcare practitioners, shaman, social institutions	**Systems and Structures**: Organisational and systemic aspects of institutions, governments, and economies that impact on experience of pain and pain services
**Ethics and Morals**: Towards suffering, care, treatment, support	** Ecosystems**: Environmental and ecological systems of modern society affecting lifestyle
**Worldviews**: Medical, social, health, religious perspectives, and narratives	**Social Systems:** Patterns and structures within societies, including cultural norms and social institutions
**Intersubjective Meaning**: Magical, mystical, mythical, divinity, medical, spiritual	**Natural environment:** Climate, weather, atmosphere
**Practitioners:** Culturally competent healthcare practitioners, social workers and counsellors, spiritual support practitioners, ethnomedicine practitioners, health coaches	**Practitioners:** Occupational therapists, social prescribers, community workers, health coaches
**Interventions (treatment/therapeutic):** Pain education, group activities/therapy, community-based support	**Interventions (treatment/therapeutic):** Policies to reduce health inequality, asset building, upstream health promotion

**Table 3 behavsci-15-00703-t003:** Mapping pain into levels of evolving worldview. Based on the author’s domain knowledge and the author’s interpretation of Wilber’s AQAL framework ([239], [242]).

Level of Development	Worldview	Characteristics	Pain	Coping Strategy	Trait
Pre-Modern: Pre-rational—Archaic level	Reality is understood through basic, sensory-motor awareness primarily focused on survival needs, with little/no sense of self or identity.	Structures of consciousness are formless with limited differentiation between human and world and non-existent cultural manifestations.	Beliefs about pain tend to be formless and rudimentary, possibly not beyond learning to avoid future encounters with noxious stimuli.	Pain and its discomfort motivate escape response, protection from injuries (e.g., guarding), avoidance of future encounters, and the use of things in the natural environment that soothes pain, e.g., warmth, cold.	An individual who associates pain with noxious stimuli but attributes little meaning to pain.
Pre-Modern: Pre-rational—Magic (animistic) level	Reality is understood through magic, animistic beliefs, symbols, rituals, and oral traditions. Individuals at this level often rely on communal beliefs and shared practices.	The beginnings of concrete thinking where experiences are explained by ‘magical thinking’ where mystical/supernatural forces are attributed to objects. Individuals perceive the world as animated and interconnected with a focus on community, tradition, ritual, and spirituality.	Pain and its inability to be resolved are seen as mysterious and beyond understanding. Individuals may attribute their pain to magical or supernatural forces and attribute superstitions or spiritual explanations to it, perhaps viewing it as a punishment or a curse.	Engaging in folk remedies and support from spiritual leaders, shamans, or community rituals.	An individual who believes their pain is due to magic, supernatural forces, a curse, or bad karma, focusing on ritualistic practices for relief.
Pre-Modern: Rational-mythic level	Reality is understood by blending religious and mythological frameworks and human realms that value belonging to a community and adhering to moral values, collective beliefs, and societal norms.	Structured thinking based on myths. Individuals attribute power and a sense of identity to cultural stories and traditions such as Greek, Egyptian, and Mesopotamian mythologies, and religious narratives.	Individuals start forming a more structured understanding of their pain through stories, myths, and cultural narratives. Pain is understood within the context of personal and mythology/religious narratives. There may be a strong attachment meaning associated with the Devine and with community beliefs regarding illness and healing.	Engaging in traditional healing practices, seeking community support, or following cultural and religious narratives about suffering and recovery.	An individual who interprets their pain through religious or cultural stories, viewing it as a test of faith or a rite of passage.
Modern: Rational-scientific level (individualist)	Reality is understood through rational analysis that values progress and innovation supported by scientific data questioning traditional beliefs and practices.	Rational, concrete, critical thinking, with scientific understanding and objective analysis and reasoning. Individuals attribute power to logic, scientific inquiry, and empirical evidence. Characterised by the emergence of a WEIRD worldview of individual rights.	Individuals adopt a scientific and rational approach to understanding their pain through a medicalised lens, focusing on diagnosis, treatment options, and biological mechanisms with an expectation that pain can be cured, fixed, and managed using biomedical treatments.	Seeking medical advice, utilising evidence-based tissue-centric treatment, the resolves pathology, with less focus on psychosocial factors or interventions. Hoping to return to pre-pain normality.	An individual who has a biomedically dominant mindset, seeking specialists, diagnostic tests, and medical treatment plans grounded in pathophysiology. May be searching for a single-treatment quick-fix cure and feeling that pain is a personal burden to a meaningful life.
Post-Modern: Rational-pluralist level	Reality is understood by valuing social justice, environmental concerns, and the importance of context in understanding truth. There is a focus on community and shared values, but often with a critique of established structures.	Rational and critical thinking that values diversity, equality, and inclusion, characterised by pluralism and relativism. Recognises the diversity of perspectives and the subjective nature of reality and engages in critical discourse around cultural and social issues.	Individuals understand pain as a multifaceted experience that can be influenced by physical, emotional, social, and cultural factors (biopsychosocial). Individuals recognise the complexity of experiences and value multiple perspectives regarding pain.	Engaging in self-education about their condition and integrative approaches that combine medical treatments with complementary and/or alternative therapies from multidisciplinary practitioners (e.g., acupuncturists, psychotherapists). May also be seeking community-based support services.	An individual who has a flexible mindset and follows a biopsychosocial approach to pain management. They are prepared to participate in support groups, explore various therapeutic modalities, and value both medical and holistic treatments.
Integral: Holistic level	Reality is understood through unity, collaboration and synthesis across different disciplines, cultures, and worldviews that value and acknowledge the complexity and diversity of human experience.	Integrated thinking that combines knowledge and perspectives from various levels of development to create a holistic understanding of reality. Power is attributed to an understanding that truth can be found in multiple forms, and that growth involves transcending and including earlier perspectives.	Pain is seen as an integral part of life that offers opportunities for growth, transformation, and deeper self-understanding. Individuals synthesise previous levels, incorporating insights and wisdom to develop a holistic and comprehensive understanding of their experience of pain.	Empowered with personal agency to engage with holistic health strategies that include biopsychosocial-spiritual approaches that Foster connections with self, others, and the environment.	An individual who has learned to accept pain as a catalyst for psychological growth, engaging in ‘spiritual’ practices, and biopsychosocial approaches to alleviate suffering; actively participates in advocacy for others with similar experiences.
Transpersonal: Transrational	Reality is understood through unity, compassion, and spiritual awareness of knowledge, experience, data, perspective, and insight to transcend the rational mind, personal self, and traditional cultural boundaries, integrating insights from various Western and Eastern traditions.	Focuses development beyond the rational mind and personal self by incorporating spiritual, mystical, and ego, self-, and spiritual transcendence experiences of connection to something greater than themselves. Values subjective experiences as sources of knowledge and insight, integrates intuition with analytical thinking and reason, and accepts contradictions and paradoxes as natural parts of complex problems. Cognitive flexibility and dialectical thought to form comprehensive understanding of complexity of situations.	Pain is seen as a state of consciousness to impart deeper meaning and purpose within a holistic life journey of discovery that transcends the personal self. Integrates biopsychosocial with spiritual, mystical, and transcendent pain experiences into holistic growth of being, belonging and becoming.	A journey of curious exploration and meaning-making of the purpose and experiences of ‘*being*’ and ‘*becoming*’ embedded in an external world (cosmos). Achieves meaning and comfort by engaging in spiritual practices (e.g., meditation, prayer) to build connection of inner-self with people, nature (other forms of life) and natural life-force/higher power.	An individual who has a deep sense of acceptance and peace with pain, who curiously explores their experiences and understandings of pain. They see pain as a constructive part of a meaningful life journey, and explores, detaches from, and relieves pain through Eastern and Western approaches.

**Table 4 behavsci-15-00703-t004:** Pain and the overarching trajectory of psychological development. Based on the author’s domain knowledge and the author’s interpretation of Wilber’s AQAL framework ([237], [238], [239], [242]). Developmental arcs imply the overall trajectory through levels of development, such as challenges, growth, and transformations experienced as an individual moves from one level to another, i.e., a journey through the levels.

Level	Characteristics	Experience of Pain	Coping	Example
Pre-personal (undifferentiated) level	This level focusses on early phases of consciousness development, typically seen in infancy and early childhood where the individual has no separate identity. This starts with a primal relationship with a collective unconscious and grows into a conscious imaginal and emotional life and the formation of mental structures and basic schemas to make sense of the world	Pain is primarily experienced in a sensory and emotional context. Infants and young children may not have the cognitive ability to understand or articulate their pain, experiencing it as a direct physical sensation or emotional distress.	Largely dependent on external support (caregivers) for comfort through physical touch, nurturing, and creating a safe environment, developing into simple self-soothing techniques, such as thumb-sucking or holding a comfort object.	An individual (infant) who is deeply connected to a caregiver (e.g., mother) and explores their surroundings through basic sensory experiences (touching, tasting objects). They exhibit reflex withdrawal from noxious stimuli and express disturbances of inner state by primal reactions (crying) and understand the world by simple schemas (mother ‘disappears’ when not in sight)
Personal (individualism) level	This level focuses on the individual’s identity, selfhood and personal experiences, including embodied emotions and physical sensations.	The individual recognises pain primarily affecting their personal sense of self and daily functioning—i.e., pain is about ‘me’. Commonly, and especially in modern and post-modern levels of development, the individual may see pain as burden or obstacle, feeling that they are victimised by their pain.	Reliance on medical treatments, therapy, and self-care strategies that focus on alleviating pain for personal relief using individualised interventions, such as psychological approaches (e.g., CBT, ACT, mindfulness) to address personal distress.	An individual with pain may view their experience primarily in terms of suffering and seek to control or eliminate the pain through conventional medical and psychological interventions that have ‘grown’ within a modern scientific [positivist] framework.
Transpersonal (interconnectedness) level	At this level, individuals start to transcend their personal identity and develop a broader awareness that includes interconnectedness with others and the external environment (e.g., nature, universe, cosmos).	The individual recognises pain as one aspect of a larger human experience that acknowledges the interconnectedness of being and becoming and understanding the connectedness and shared nature of such experiences, including suffering, within the context of ‘things’ that exist. Pain is seen as an opportunity for curious exploration of the shared and interconnected nature of this experience with others who may endure similar experiences. This may still be within a personal construct that desires to relieve ‘my pain and suffering’.	Strategies, techniques, and interventions that encompass a wide range of eclectic practices both within and beyond the conventional biopsychosocial healthcare domain. These may include support groups, community involvement, volunteering, and exploring spiritual practices, meditation, or contemplative approaches to gain deeper insights from pain experiences. Somatic practices like yoga, Reiki, and traditional Chinese acupuncture focus on ‘releasing energy or emotional blockages’, irrespective of whether this explanation is considered literal or metaphorical. Additionally, deep meditation, mindfulness, and transcendental experiences can foster a sense of unity and acceptance of suffering. Practices emphasising non-duality and oneness, such as advanced spiritual inquiry and contemplative awareness, help individuals cultivate compassion for themselves and others as part of human experience.	An individual finds meaning in their pain by connecting with themselves (e.g., through introspection) and connecting with others (e.g., through engagement with pain communities), fostering a sense of empathy and compassion towards others who suffer.

**Table 5 behavsci-15-00703-t005:** Features of levels of development and states of consciousness for pain. Based on the author’s domain knowledge and the author’s interpretation of Wilber’s AQAL framework ([239], [242]).

Aspect	Levels of Development	States of Consciousness
Definition	The progressive levels of understanding and coping that individuals with long-term pain may experience over time.	The transient/temporary conditions or experiences of awareness that individuals access whilst in pain.
Characteristics	Progressive and stable, representing long-term growth and reflecting evolving emotional and cognitive development. Individuals move through an evolving relationship with pain to develop more sophisticated understandings and strategies for improved coping and resilience.	Dynamic and fluctuating states of awareness that may be fleeting or prolonged and are influenced by circumstance and impact on immediate experiences of pain and respond to it.
Examples	Levels: Egocentric, ethnocentric, world-centric, cosmo-centric; or pre-modern, modern, post-modern, integral; or pre-personal, personal, transpersonal. Lines: Cognitive; moral; emotional; interpersonal; spiritual; aesthetic; physical.	Coma, asleep, awake relaxed, awake alert, aroused, delirious, psychotic, altered. Pain: Fluctuating intensity (heightened, diminished), fluctuating location, fluctuating quality, movement-evoked pain, breakthrough pain, flare-up.
Implication	Guides long-term treatment strategies and personal growth.	Affects day-to-day coping and quality of life.
Example	Pre-rational: A magical or mythic understanding of pain, attributing it to supernatural causes or viewing it as a punishment. Rational and individualistic: A logical, scientific understanding of pain related to tissue damage that requires medical explanation and treatment. Rational and pluralistic: A biopsychosocial understanding of pain with a broader understanding of pain and its impact requiring multimodal strategies, including mindfulness or community support. Integral/Transrational: Pain becomes a catalyst for spiritual growth or profound insights about life and suffering.	Acute state: During an injury or a flare when there is heightened awareness of pain, leading to feelings of despair, anxiety, or frustration. Distracted state: In moments of distraction or engagement in enjoyable activities, individuals may enter a state where their pain feels less significant or overwhelming. Altered state: Mindfulness or meditation help individuals to detach from their pain allowing for a less reactive relationship with their experience.
Implication	Guides long-term treatment strategies and personal growth.	Affects day-to-day coping and quality of life.

## Data Availability

Underlying materials related to this manuscript can be accessed by contacting Professor Mark I. Johnson.

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
