# Peer review of "Reconfiguring Pain Interpretation Within a Social Model of Health Using a Simplified Version of Wilber’s All Quadrant All Levels Framework: An Integral Vision"

_behavsci, 2025, doi:10.3390/bs15050703_

Round 1

Reviewer 1 Report

Comments and Suggestions for Authors

Thank you for the opportunity to read this really complex piece but well thought out piece of work which challenges a paradigm widely promoted and perhaps not always far removed from the biomedical paradigm. I would like to see some more suggestions of how the concepts relate to a clinical setting and could be operationalised. Please see below for some suggestions.

Line 102, quotation marks for ‘dis-ease’ would be helpful to highlight this insightful term rather than it looking like a typo.

Okolo’s reference is good but an example from this paper or a few others would really help to contextualise how this is done. 

Line 171 some thing afoot with the referencing.

Table 3 spell out VCSE here at first reference please

806-817 - would this be better placed in the introduction where you introduce these terms.

976 - a clear definition of this interesting term, which is central to this paper ‘painogenic’ would be valuable.

Some examples of how Rethinking Pain works, operationalises this idea of vcse involvement in health care would be really helpful. 

Author Response

I would like to express sincere gratitude to the reviewers for taking the time to review my manuscript and provide constructive comments. I have taken on board, and accepted all points raised and we believe that this has markedly improved the quality of my manuscript.

The nature and extent of the suggested amendments, including re organisation of content make tracking changes challenging.

My responses to each point raised by reviewers are provided below. I have uploaded the following manuscripts document to assist tracking of changes

  • a clean unmarked version
    • Filename: behavsci-3553525_REVISED_CLEAN_10-04-2025
  • a clean unmarked version with responses to the reviewers’ main suggestions as ‘comments’
    • Filename: behavsci-3553525_REVISED_COMMENT-TRACKING_10-04-2025
  • a marked-up ‘track change’ version of the manuscript highlighting gross amendments and responses to all of reviewers’ suggestions as ‘comments’
    • Filename: behavsci-3553525(R1)_TRACKCHANGE_08-04-2025_LOCKED

I am aware of some typographical inconsistencies in the in-text reference citations that will be attended to if the manuscript is accepted for publication.

I sincerely hope my revised manuscript meets with your approval as I believe it will be of great interest to the readers of Behavioural Sciences.

Kindest regards

Prof. Mark I. Johnson

Reviewer 1

Thank you for the opportunity to read this really complex piece but well thought out piece of work which challenges a paradigm widely promoted and perhaps not always far removed from the biomedical paradigm. I would like to see some more suggestions of how the concepts relate to a clinical setting and could be operationalised. Please see below for some suggestions.

Line 102, quotation marks for ‘dis-ease’ would be helpful to highlight this insightful term rather than it looking like a typo.

Okolo’s reference is good but an example from this paper or a few others would really help to contextualise how this is done. 

Author’s Response: Amended

Line 171 some thing afoot with the referencing.

Author’s Response: Amended

Table 3 spell out VCSE here at first reference please

Author’s Response: Amended

806-817 - would this be better placed in the introduction where you introduce these terms.

Author’s Response: Amended

976 - a clear definition of this interesting term, which is central to this paper ‘painogenic’ would be valuable.

Author’s Response: Amended

Some examples of how Rethinking Pain works, operationalises this idea of vcse involvement in health care would be really helpful. 

Author’s Response: Amended. I have expanded this section and of have developed discussions of how integral pain relates to clinical (healthcare) settings and can be operationalised.

Reviewer 2 Report

Comments and Suggestions for Authors

This submission is an ambitious project from several perspectives. One is that the author is taking a theory by a popular psychology author that has not been examined in peer-reviewed journals and introducing it in an academic journal submission to change academic perceptions of long-term pain. Another is that the author presents this popular theory as able to revolutionize the interpretation and understanding of long-term pain. A third is that the author offers an account significantly longer than most journal articles—meaning that the author intends to change minds in a way that is less likely to be attractive to academic journal article readers.

It is evident from the article that the author has an extensive grasp of the issue of long-term pain management based on substantial thought and interpretation. This reach and insight in putting together a point of view by the author is commendable. As such, it is worthy of consideration.

With all that is admirable about this undertaking and its results, there are problems regarding how the work is structured, the citation method, unnecessary point repetition, and too many outdated references. Highlighting these problems is a line-by-line assessment of the work.

Line by line suggested edits

1 This work is presented as a Review. However, it is not a review. It is a point of view based on a theory by one writer presented in several books. Please rename this as a Viewpoint.

12-30 The Abstract structure should follow the standardized form: Background, Methods, Results, and Conclusion. This Abstract is entirely Background. Please rewrite it to follow the accepted standard and keep the word count within the 200-word limit.

31-32 Keywords must be in the Abstract. Either include these current keywords in the Abstract or select ones found in the Abstract.

35 It is unusual in a scientific journal to have quotations by various authors interspersed throughout the manuscript. These quotations distract the reader from the main text and may distort the argument presented if they refocus the reader. It is best to eliminate the quotation in this line and all that follow.

38-42 After deleting the quotation, please delete these lines.

42-57 There are no citations in these two paragraphs. However, the author makes claims in each sentence. All claims require citation support. The gold standard in scientific research is citations to publications from the previous five years (since 2021).

58-62 These two sentences require supporting citations.

70-78 The citations in this paragraph are all outdated. Please find supporting citations to research published since 2021.

88-89 The author cites three publications by Wilber, yet the reference list includes ten. Please cite all publications mentioning Integral Theory by Wilber.

90 Once the author has admitted that this theory has not been peer-reviewed, the author must present why its consideration should be serious—something not mentioned until lines 1062-1067. Please move the information in lines 1062-1067 to the Introduction section, explaining why it is reasonable to consider the work of Wilber.  Move lines 986-993 to the Introduction section to provide an additional reason for considering the Integral Theory sensible, and move lines 655-704 so that its strengths, weaknesses, and criticisms are there too. The author must present a strong case for why this non-academic theory should be the basis of change considering long-term pain.

96 Please eliminate this point. Defining WEIRD must be in the Introduction, not mentioned as a result.

102-107 These lines are the only ones that hint at the methodology used in the analysis of the work of Wilber on Integral Theory. Section 2 should be the Methods section. A description of the methodological analysis of Integral Theory must be in the Methods section.

108-131 Please delete these lines. These topics are too large and intricate for such a cursory examination. It is better not to mention this history at all.

132-218 Please move these lines to be part of the Introduction—eliminating all the quotations.

219 Please delete this heading.

220-253 Please move this information to the new aspect of the Introduction explaining why adopting Wilber’s Integral Theory is reasonable.

254-255 Please delete these headings.

256-282 Please shorten this information on Wilber and move it directly after line 92.

283 This is where the section “2. Methods” should begin. The author must describe the methodology used to assess Wilber’s Integral Theory and provide relevant citations to research published since 2021 to demonstrate the currency of the methodology in pain research.

284-1029. When reworked, these lines represent section “3. Results”. Expected changes include the following.

  1. Eliminate all the quotations.
  2. Provide citations to Wilber for each of the claims made in this Results section.
  3. Ensure that all tables and figures are mentioned in the text before they appear.   
  4. Increase the size of the figures so that the information is legible.
  5. Reduce the wording of the figure captions for Figure 1, Figure  2, Figure 7, Figure 8, Figure 9, and Figure 10, providing only essential information to understand the figures.
  6. Please eliminate the line regarding theorists in Table 1. This information does not do the theorists justice since the author does not discuss each theorist in detail in the text.
  7. Table 3 adds very little to the information already in UL Table 2. Please eliminate it.
  8. Table 4 adds very little to the information already in UR Table 2. Please eliminate it.
  9. Table 5 adds very little to the information already in LL Table 2. Please eliminate it.
  10. Table 6 adds very little to the information already in LR Table 2. Please eliminate it.

649-The point of the Discussion in the case of presenting a non-academic theory to an academic audience is to examine other competing theories to that of Wilber regarding their suitability for making progressive changes to the understanding of the treatment of long-term pain. Instead, the author uses the Discussion to expand on Wilber’s theory. The Discussion can only expand upon the presented theory in contrast to the weaknesses of other theories.

1085 The Discussion must end with a paragraph regarding the limitations of the author’s assessment of Wilber’s theory.

1087-1109 These future directions must relate to suggested future research directions. Please indicate the type of research necessary to accomplish the presented future directions.

1119-1120 Please delete “as” and the quotation that follows.

1121-1125 Delete this section. There is only one author.

1134-1138 This information regarding the use of AI in assisting in the creation of this manuscript and the generation of the figures is very unusual. It will be up to the editors to decide if the use of AI is acceptable. For most MDPI journals, the use of AI  is unacceptable.

Author Response

I would like to express sincere gratitude to the reviewers for taking the time to review my manuscript and provide constructive comments. I have taken on board, and accepted all points raised and we believe that this has markedly improved the quality of my manuscript.

The nature and extent of the suggested amendments, including re organisation of content make tracking changes challenging.

My responses to each point raised by reviewers are provided below. I have uploaded the following manuscripts document to assist tracking of changes

  • a clean unmarked version
    • Filename: behavsci-3553525_REVISED_CLEAN_10-04-2025
  • a clean unmarked version with responses to the reviewers’ main suggestions as ‘comments’
    • Filename: behavsci-3553525_REVISED_COMMENT-TRACKING_10-04-2025
  • a marked-up ‘track change’ version of the manuscript highlighting gross amendments and responses to all of reviewers’ suggestions as ‘comments’
    • Filename: behavsci-3553525(R1)_TRACKCHANGE_08-04-2025_LOCKED

I am aware of some typographical inconsistencies in the in-text reference citations that will be attended to if the manuscript is accepted for publication.

I sincerely hope my revised manuscript meets with your approval as I believe it will be of great interest to the readers of Behavioural Sciences.

Kindest regards

Prof. Mark I. Johnson

Reviewer 2

This submission is an ambitious project from several perspectives. One is that the author is taking a theory by a popular psychology author that has not been examined in peer-reviewed journals and introducing it in an academic journal submission to change academic perceptions of long-term pain. Another is that the author presents this popular theory as able to revolutionize the interpretation and understanding of long-term pain. A third is that the author offers an account significantly longer than most journal articles—meaning that the author intends to change minds in a way that is less likely to be attractive to academic journal article readers.

It is evident from the article that the author has an extensive grasp of the issue of long-term pain management based on substantial thought and interpretation. This reach and insight in putting together a point of view by the author is commendable. As such, it is worthy of consideration.

With all that is admirable about this undertaking and its results, there are problems regarding how the work is structured, the citation method, unnecessary point repetition, and too many outdated references. Highlighting these problems is a line-by-line assessment of the work.

Author’s Response: I am extremely grateful for your insight, and this has really helped me to see how communication of the work could be improved. I agree that a stronger case for using the AQAL framework was needed and the methodological process needed to be more visible.  I have spent considerable time attempting to do this and feel that this has sharpened the manuscript. Responses to each suggestion is provided below.

Line by line suggested edits

1 This work is presented as a Review. However, it is not a review. It is a point of view based on a theory by one writer presented in several books. Please rename this as a Viewpoint.

Author’s Response: Amended

12-30 The Abstract structure should follow the standardized form: Background, Methods, Results, and Conclusion. This Abstract is entirely Background. Please rewrite it to follow the accepted standard and keep the word count within the 200-word limit.

Author’s Response: Amended

31-32 Keywords must be in the Abstract. Either include these current keywords in the Abstract or select ones found in the Abstract.

Author’s Response: Amended

35 It is unusual in a scientific journal to have quotations by various authors interspersed throughout the manuscript. These quotations distract the reader from the main text and may distort the argument presented if they refocus the reader. It is best to eliminate the quotation in this line and all that follow.

Author’s Response: Amended – all quotes removed.

38-42 After deleting the quotation, please delete these lines.

Author’s Response: Amended – all quotes removed.

42-57 There are no citations in these two paragraphs. However, the author makes claims in each sentence. All claims require citation support. The gold standard in scientific research is citations to publications from the previous five years (since 2021).

Author’s Response: Amended by adding reference citations of up-to-date evidence through all sections of the manuscript. Citations of some less recent seminal works remain.

58-62 These two sentences require supporting citations.

Author’s Response: Amended – supporting citations added throughout the manuscript.

70-78 The citations in this paragraph are all outdated. Please find supporting citations to research published since 2021.

Author’s Response: Amended – supporting citations of up-to-date evidence added throughout the manuscript.

88-89 The author cites three publications by Wilber, yet the reference list includes ten. Please cite all publications mentioning Integral Theory by Wilber.

Author’s Response: Amended

90 Once the author has admitted that this theory has not been peer-reviewed, the author must present why its consideration should be serious—something not mentioned until lines 1062-1067. Please move the information in lines 1062-1067 to the Introduction section, explaining why it is reasonable to consider the work of Wilber.  Move lines 986-993 to the Introduction section to provide an additional reason for considering the Integral Theory sensible, and move lines 655-704 so that its strengths, weaknesses, and criticisms are there too. The author must present a strong case for why this non-academic theory should be the basis of change considering long-term pain.

Author’s Response: Amended – I have made explicit, in the Introduction, that even though Wilber’s Integral Theory of consciousness has not been validated, the application of a of Wilber’s AQAL framework as a simple heuristic is appropriate to advance understanding – lines 83-111. Consequences of these shortcomings are discussed in 4.4 Limitations lines 1022-1044. I have balanced the tone of the Discussion so as not to overplay the implications of the findings. 

96 Please eliminate this point. Defining WEIRD must be in the Introduction, not mentioned as a result.

Author’s Response: Amended.

102-107 These lines are the only ones that hint at the methodology used in the analysis of the work of Wilber on Integral Theory. Section 2 should be the Methods section. A description of the methodological analysis of Integral Theory must be in the Methods section.

Author’s Response: Amended. I believe this markedly improves reporting of the methodological process undertaken – thank you.

108-131 Please delete these lines. These topics are too large and intricate for such a cursory examination. It is better not to mention this history at all.

Author’s Response: Amended – section deleted

132-218 Please move these lines to be part of the Introduction—eliminating all the quotations.

Author’s Response: Amended

219 Please delete this heading.

Author’s Response: Amended

220-253 Please move this information to the new aspect of the Introduction explaining why adopting Wilber’s Integral Theory is reasonable.

Author’s Response: Amended – see previous comment

254-255 Please delete these headings.

Author’s Response: Amended

256-282 Please shorten this information on Wilber and move it directly after line 92.

Author’s Response: Amended

283 This is where the section “2. Methods” should begin. The author must describe the methodology used to assess Wilber’s Integral Theory and provide relevant citations to research published since 2021 to demonstrate the currency of the methodology in pain research.

Author’s Response: Amended. As a point of clarity I am not assessing Wilber’s Integral Theory – I am utilising Wilber’s AQAL as a heuristic device. I have emphasised this point in the manuscript, I have discussed the currency of the methodology in pain research in section 1.2 Context lines 122 -227, which lays out the need for new approaches and perspectives to advance the biopsychosocial understanding. I hope you agree that this section merits inclusion - I tried a variety of ways of shortening and repositioning this section.

284-1029. When reworked, these lines represent section “3. Results”. Expected changes include the following.

284-1029. When reworked, these lines represent section “3. Results”.

  1. Eliminate all the quotations. - Amended
  2. Provide citations to Wilber for each of the claims made in this Results section - Amended
  3. Ensure that all tables and figures are mentioned in the text before they appear. - Amended 
  4. Increase the size of the figures so that the information is legible.- Amended. I have also removed AI- generated images to improve the clarity of the figures.
  5. Reduce the wording of the figure captions for Figure 1, Figure  2, Figure 7, Figure 8, Figure 9, and Figure 10, providing only essential information to understand the figures. - Amended
  6. Please eliminate the line regarding theorists in Table 1. This information does not do the theorists justice since the author does not discuss each theorist in detail in the text. - Amended
  7. Table 3 adds very little to the information already in UL Table 2. Please eliminate it. - Amended
  8. Table 4 adds very little to the information already in UR Table 2. Please eliminate it. - Amended
  9. Table 5 adds very little to the information already in LL Table 2. Please eliminate it. - Amended
  10. Table 6 adds very little to the information already in LR Table 2. Please eliminate it. - Amended

Author’s Response: Amended. I have reshaped this section. I have deleted Table 10 from the Discussion and presented Tables 3-6 and 10 as Supplementary Material.

649-The point of the Discussion in the case of presenting a non-academic theory to an academic audience is to examine other competing theories to that of Wilber regarding their suitability for making progressive changes to the understanding of the treatment of long-term pain. Instead, the author uses the Discussion to expand on Wilber’s theory. The Discussion can only expand upon the presented theory in contrast to the weaknesses of other theories.

Author’s Response: Amended. I have reshaped the Discussion section and restructured other sections based on your comments to improve the communication of the implications of the findings.  It was not my intention to appraise Wilber’s Integral Theory of Everything nor to create an explanatory model of pain. Rather the intention was to use the AQAL framework to represent the biopsychosocial model through objective, subjective, individual and collective perspectives and psychological growth. I hope this is clearer now.

1085 The Discussion must end with a paragraph regarding the limitations of the author’s assessment of Wilber’s theory.

Author’s Response: Amended

1087-1109 These future directions must relate to suggested future research directions. Please indicate the type of research necessary to accomplish the presented future directions.

Author’s Response: Amended

1119-1120 Please delete “as” and the quotation that follows.

Author’s Response: Amended

1121-1125 Delete this section. There is only one author.

Author’s Response: Amended

1134-1138 This information regarding the use of AI in assisting in the creation of this manuscript and the generation of the figures is very unusual. It will be up to the editors to decide if the use of AI is acceptable. For most MDPI journals, the use of AI  is unacceptable.

Author’s Response: I have removed all AI generated images from figures. I have used GenAI very occasionally for the purposes of text editing (e.g., grammar, structure, spelling, punctuation and formatting). Journal policy is a little unclear about what needs to be declared. i.e., “AI technology can still be used when writing academic papers. However, this must be appropriately declared when submitting a paper to an MDPI journal. In such cases, authors are required to be fully transparent, within the “Acknowledgments” section, about which tools were used, and to describe in detail how the tools were used, in the “Materials and Methods” section.” … but also … “The use of GenAI tools for the purposes of text editing (e.g., grammar, structure, spelling, punctuation and formatting) is not covered by this policy and does not need to be declared.” 

I always err on the side of caution and disclose everything in the spirit of transparency. I have used MDPI’s recommended AI acknowledgement statement and will leave it up to the journal editors whether this statement should remain.

Round 2

Reviewer 2 Report

Comments and Suggestions for Authors

Thank you to the author for the changes made to the manuscript. All have improved it. However, there are still changes remaining.

The author has a vast knowledge of this subject. The intent is to change the thinking of colleagues. To do so, the author must balance the amount he knows with what may convince others. The following comments are from this perspective. If the author has made a change without a comment by the reviewer, the change is accepted.

Line by line suggested edits regarding the clean copy
55 Please find support for the citation to a reference published since 2021.
69 The information on the subsection 1.2. Context should come here—after introducing the biopsychosocial model.
110-120 The Aim comes too early in this Introduction. It should be the last section of the Introduction.
134 Please cite this "series of books".
138-139 Please cite the “large body of empirical research”.
143 The author must provide evidence that the AOAL framework is unique by presenting the differences with other similar frameworks.
160-161 Please delete this sentence. 
163 Please find support for the citation to a reference published since 2021.
181 Please delete “perpetuates Cartesian dualism and”—this is not a philosophy journal.
204-228 Although these paragraphs provide interesting and relevant information for a more detailed discussion, they are distracting regarding this viewpoint. Please delete these lines.
229-301 This Materials and Methods section concerns those of Wilber, not the materials and methods the author uses to interpret his work. Please provide a paragraph regarding the methodology the author is using to interpret Wilber. Mention of all tables and figures must be in the text before presenting the tables and figures. Here, in these paragraphs preceding the tables and figures, the author should use a stated methodology to interpret them. 
302-320 This section on "Research Question" belongs as part of the Aim of the study. It does not belong in the Materials and Methods. Please add this information to the Aim subsection in the Introduction.
321-749 Please be precise in the Results that these are the results of interpreting Wilber. Generally, the journal prefers sections divided into, at the most, subsubsections. These results have subsubsubsections—this is likely unacceptable to the journal. Please rework the results to eliminate the subsubsubsections and clarify how these divisions concern the interpretation the author offers. The author should aim not to replicate Wilber.
790-819 It is unusual for a Discussion section to have figures because they should be part of the Results. Please rework the Discussion to move the two figures to the Results. The Discussion should concentrate on how the findings in the results relate to previous interpretations of Wilber (if there are any). Also, the Discussion should focus on the improvement made to the current understanding of pain by the interpretation of Wilber offered. Similar to the advice for the Results section, please rewrite the Discussion to eliminate the subsubsubsections. Please include supporting citations for all citations to research published since 2021.
1022 Please cite these criticisms.
1033 Please find support for the citation to a reference published since 2021.
1042 Please find support for the 2004 citation to a reference published since 2021.
1047 The presentation of the process guiding the viewpoint is the one that must be in the Materials and Methodology. 
1050 Please provide a paragraph on future research directions.
1052 The Conclusion must return to the research questions, presenting their answers. 

Author Response

Response to Reviewer – R2

behavsci-3553525

22 April 2025

Once again, I would like to express sincere gratitude to reviewer 2 for taking time to offer constructive comments about my manuscript. I accept all suggestions for improvement and have made substantive amendments to all sections of the manuscript to improve the clarity, depth and detail of communication of the study.

The nature and extent of the suggested amendments, including reorganisation of content make tracking all changes challenging.  I have uploaded a clean unmarked version (Filename: behavsci-3553525_clean version_REVISED_Round2_CLEAN_22-04-2025_SUBMITTED) and a clean unmarked version with ‘comments’ that also includes an appendix of gross level track changes (Filename: behavsci-3553525_clean version_REVISED_Round2_Comments_22-04-2025). I emphasise that track changes only highlight the extent of gross amendments, and that substantive proof editing was undertaken post track change.

My responses to each point raised by the reviewer with line numbers are provided below. I am aware that there may be some typographical inconsistencies in the in-text reference citations that will be attended to if the manuscript is accepted for publication.

I hope my revised manuscript meets with your approval.

Kindest regards

Prof. Mark I. Johnson

Reviewer 2 comments

Thank you to the author for the changes made to the manuscript. All have improved it. However, there are still changes remaining.

The author has a vast knowledge of this subject. The intent is to change the thinking of colleagues. To do so, the author must balance the amount he knows with what may convince others. The following comments are from this perspective.

 If the author has made a change without a comment by the reviewer, the change is accepted. Line by line suggested edits regarding the clean copy

Response to comments

55 Please find support for the citation to a reference published since 2021.
Author’s response: Added

 69 The information on the subsection 1.2. Context should come here—after introducing the biopsychosocial model.
Author’s response: Amended. I have reworked the flow of information in the Introduction including the addition of supporting evidence.

110-120 The Aim comes too early in this Introduction. It should be the last section of the Introduction.
Author’s response: Amended. Lines 220 onwards

134 Please cite this "series of books".

Author’s response: Do you mean line 92? – Amended. Lines 167-170.

138-139 Please cite the “large body of empirical research”.
Author’s response: Amended. I have also qualified the ‘status’ of the WEIRD concept. Lines 92-114.

143 The author must provide evidence that the AOAL framework is unique by presenting the differences with other similar frameworks.
Author’s response: Do you mean line 99? – Amended by comparing with other frameworks and supported by evidence. Lines 188-229

160-161 Please delete this sentence. 
Author’s response: Amended.

163 Please find support for the citation to a reference published since 2021.

Author’s response: Amended.

181 Please delete “perpetuates Cartesian dualism and”—this is not a philosophy journal.
Author’s response: Amended.

204-228 Although these paragraphs provide interesting and relevant information for a more detailed discussion, they are distracting regarding this viewpoint. Please delete these lines.

Author’s response: Deleted from the reworked Introduction.
229-301 This Materials and Methods section concerns those of Wilber, not the materials and methods the author uses to interpret his work. Please provide a paragraph regarding the methodology the author is using to interpret Wilber. Mention of all tables and figures must be in the text before presenting the tables and figures. Here, in these paragraphs preceding the tables and figures, the author should use a stated methodology to interpret them. 
Author’s response: I have added a clearer description of my approach at the beginning of the Methods section and have contextualised figures and tables as suggested. Lines 251-265, with an outline to the AQAL framework from lines 268-329.

302-320 This section on "Research Question" belongs as part of the Aim of the study. It does not belong in the Materials and Methods. Please add this information to the Aim subsection in the Introduction.
Author’s response: Amended. Lines 220 onwards

321-749 Please be precise in the Results that these are the results of interpreting Wilber. Generally, the journal prefers sections divided into, at the most, subsubsections. These results have subsubsubsections—this is likely unacceptable to the journal. Please rework the results to eliminate the subsubsubsections and clarify how these divisions concern the interpretation the author offers. The author should aim not to replicate Wilber.

Author’s response: Amended. I have reworked the Results to emphasise that these are the results of applying Wilber’s AQAL framework and importantly Wilber’s levels of development. I am not to replicating Wilber as Wilber nor anyone else has used an AQAL framework to map pain. I have relocated and integrated tables, figures and associated narrative from the Discussion into the Results. In the text-narrative I refer to research literature to support the mapping process. The journal publishes level 3 sub-headings. I have removed the majority of level 3 sub-headings and all of the level 4 subheadings, and contextualised the information accordingly. Lines 329-789.

790-819 It is unusual for a Discussion section to have figures because they should be part of the Results. Please rework the Discussion to move the two figures to the Results. The Discussion should concentrate on how the findings in the results relate to previous interpretations of Wilber (if there are any). Also, the Discussion should focus on the improvement made to the current understanding of pain by the interpretation of Wilber offered. Similar to the advice for the Results section, please rewrite the Discussion to eliminate the subsubsubsections. Please include supporting citations for all citations to research published since 2021.

Author’s response: Amended. I have moved the figures to the Results and reworked the Discussion section in line with the comments above, including updating citations to research published since 2021.  This has involved more focussed alignment of the insights and implications of the findings of the mapping exercise with objectives 1-3.  I have removed all of the level 4 subsections but have left in some level 3 sub-headings to assist the reader. Lines 772-1104.

1022 Please cite these criticisms.

Author’s response: Amended. Lines 1033-1035.
1033 Please find support for the citation to a reference published since 2021.

Author’s response: Amended. Lines 1051-1052.
1042 Please find support for the 2004 citation to a reference published since 2021.
Author’s response: Amended. Lines 1059-1060.

1047 The presentation of the process guiding the viewpoint is the one that must be in the Materials and Methodology. 
Author’s response: Amended. This comment has been reframed to emphasise the limitation of methodological approach. Lines 1065-1067.

1050 Please provide a paragraph on future research directions.

Author’s response: Amended. Lines 1073-1103.
1052 The Conclusion must return to the research questions, presenting their answers. .

Author’s response: Amended. Conclusion has been reworked. Lines 1107-onwards

Round 3

Reviewer 2 Report

Comments and Suggestions for Authors

Thank you to the author for the changes made. They are substantial, and all have improved the submission. A few points remain.

Now that clarity is provided that the author is presenting new research, not just a viewpoint, please change the publication type from Viewpoint to Article.

The manuscript Title must be more reflective of the contribution article. Please change the title to “A Simplified Version of Wilber’s All Quadrant All Levels Framework to Provide a Reconfiguration of Pain Interpretation Within a Social Model of Health”.

Please rework the Abstract to explain that the method is narrative research methodology regarding Ken Wilber’s AQAL theory—it is not Wilber’s theory itself.

Lines 305-339 belong in the Discussion section. In the Discussion, the author presents these alternative frameworks and, based on the simplified version created in this manuscript, replies to why the simplified version of AQAL is more effective regarding pain interpretation.

355 After this line, please present the hypothesis regarding the aim.

362-374 These lines attempt to provide the context for the methodology. If the author hopes to convince readers of the value of this contribution, the methodology requires additional grounding. Here are some references concerning narrative research methodology that may be helpful to the author in providing this grounding. DOI: 10.46743/2160-3715/2021.5010, http://content.apa.org/books/17220-000, 10.1177/10497323231158619.

377-514 In this Materials and Methods section, the author has not made it sufficiently clear for the figures that this is the author’s interpretation of Wilber in presenting this simplified approach. This clarity must be in the text and the captions for the figures.

516 When beginning this Results section, please clarify that this extraction is the author’s interpretation of Wilber based on narrative research methodology.

1432 Please add that the method used is narrative research methodology and cite the same references the author has selected for lines 362-374 above.

1434 The author must cite research on cognitive bias, such as DOI: 10.1037/lhb0000482, DOI: 10.1162/posc_a_00589, and DOI: 10.1016/j.ipm.2024.103672, indicating the steps the author took to avoid cognitive bias.

1475 Once there is a hypothesis in the Introduction, the author must return to if there is confirmation of the hypothesis in the Conclusion.

Author Response

Response to Reviewers – R3 behavsci-3553525

01 May 2025

Thank you to Reviewer 2 for further suggestions for improvement of the manuscript. I have reworked sections of the manuscript as reflected in my responses below. I have uploaded a marked-up ‘track change’ version of the manuscript but emphasise that this only highlights initial gross amendments, and that additional editing was undertaken post track change. I am aware that there may be some typographical inconsistencies in the in-text reference citations that will be attended to if the manuscript is accepted for publication. I will also undertake another thorough proof edit. I hope my revised manuscript meets with your approval.

I am most grateful for the constructive advice throughout the review process.

Kindest regards, Prof. Mark I. Johnson

Reviewer 2 comments

Thank you to the author for the changes made. They are substantial, and all have improved the submission. A few points remain.

Response to comments

Now that clarity is provided that the author is presenting new research, not just a viewpoint, please change the publication type from Viewpoint to Article.

Author’s response: Amended. Line 1.

The manuscript Title must be more reflective of the contribution article. Please change the title to “A Simplified Version of Wilber’s All Quadrant All Levels Framework to Provide a Reconfiguration of Pain Interpretation Within a Social Model of Health”.
Author’s response: I agree that the title needs amending. I would like to use the following title: Reconfiguring Pain Interpretation Within a Social Model of Health Using a Simplified Version of Wilber’s All Quadrant All Levels Framework: An Integral Vision. However, if this is not acceptable I will revert to the one suggested by the reviewer. Amended. Line 2.

Please rework the Abstract to explain that the method is narrative research methodology regarding Ken Wilber’s AQAL theory—it is not Wilber’s theory itself.

Author’s response: Amended. Line 13.

Lines 305-339 belong in the Discussion section. In the Discussion, the author presents these alternative frameworks and, based on the simplified version created in this manuscript, replies to why the simplified version of AQAL is more effective regarding pain interpretation.

Author’s response: Amended. Moved and integrated into Discussion. Line 815 

355 After this line, please present the hypothesis regarding the aim.

Author’s response: Amended. Line 210.

362-374 These lines attempt to provide the context for the methodology. If the author hopes to convince readers of the value of this contribution, the methodology requires additional grounding. Here are some references concerning narrative research methodology that may be helpful to the author in providing this grounding. DOI: 10.46743/2160-3715/2021.5010, http://content.apa.org/books/17220-000, 10.1177/10497323231158619.

Author’s response: Amended. The methods section has been reworked [Lines 226-269] to include additional grounding supported with references and the author’s positionality [lines 249-269]. Thank you for the references – I found the Weiss & Johnson-Koenke, 2023 article fascinating.

377-514 In this Materials and Methods section, the author has not made it sufficiently clear for the figures that this is the author’s interpretation of Wilber in presenting this simplified approach. This clarity must be in the text and the captions for the figures.

Author’s response: Amended. Line 272 and continued throughout the rest of the manuscript text and table/figure captions. 

516 When beginning this Results section, please clarify that this extraction is the author’s interpretation of Wilber based on narrative research methodology.

Author’s response: Amended. Line 334.

1432 Please add that the method used is narrative research methodology and cite the same references the author has selected for lines 362-374 above.

Author’s response: Added.  Line 1132 onwards.

1434 The author must cite research on cognitive bias, such as DOI: 10.1037/lhb0000482, DOI: 10.1162/posc_a_00589, and DOI: 10.1016/j.ipm.2024.103672, indicating the steps the author took to avoid cognitive bias.

Author’s response: Amended by discussing the impact of cognitive bias on the findings and the steps to mitigate this. Line 1132 onwards

1475 Once there is a hypothesis in the Introduction, the author must return to if there is confirmation of the hypothesis in the Conclusion.

Author’s response: Amended. Line 1212.